# Inference-time Scaling of Diffusion Models through Classical Search

**Xiangcheng Zhang**[1]**, Haowei Lin**[1]**, Haotian Ye**[2]**, James Zou**[2]
**Jianzhu Ma**[1]**, Yitao Liang**[1]**, Yilun Du**[3]*
[1]Helixon US Inc.    [2]Stanford University    [3]Harvard University

## Abstract

Classical search algorithms have long underpinned modern artificial intelligence. In this work, we tackle the challenge of inference-time control in diffusion models—adapting generated outputs to meet diverse test-time objectives—using principles from classical search. We propose a general framework that orchestrates local and global search to efficiently navigate the generative space. It performs compute-efficient global exploration using breadth-first and depth-first tree search and employs a theoretically grounded, scalable local search via annealed Langevin MCMC. We evaluate our approach on a range of challenging domains, including planning, offline reinforcement learning, and image generation, and observe significant gains in both performance and efficiency over baseline methods. These results demonstrate that classical search offers a principled and practical foundation for inference-time scaling in diffusion models. By jointly scaling local and global search for the first time, our framework establishes a new Pareto frontier across challenging decision-making domains.

## 1 Introduction

Classical search algorithms have laid the foundation for modern artificial intelligence (Russell & Norvig, 2009). In discrete settings, graph search algorithms are widely used to explore the state space. Breadth-first search (BFS) (Moore, 1959) and depth-first search (DFS) (Tarjan, 1972) traverse the search tree in a fixed order. To better leverage problem-specific information, best-first search methods (Pearl, 1984), such as A* (Hart et al., 1968), use a heuristic to evaluate and prioritize states. Alternatively, local search methods, such as hill-climbing (Russell & Norvig, 2009, Sec. 4.1), explore neighboring states. More recent techniques like gradient descent and Markov Chain Monte Carlo (MCMC) have become widely adopted in optimization and probabilistic inference, underpinning many modern AI models.

Diffusion models (Ho et al., 2020) have shown impressive performance in generative modeling for continuous domains such as images (Dhariwal & Nichol, 2021), videos (Ho et al., 2022). They are also increasingly used in robotics and decision-making (Liu et al., 2024; Black et al., 2024; Team et al., 2024) to generate diverse actions (Chi et al., 2023). However, generated samples may not always align with physical laws (Song et al., 2023) or human intent (Wallace et al., 2024), and the vast generative space often necessitates multiple trials to produce satisfactory outputs (Xie et al., 2025). To address this, we scale up *inference-time compute* using strategic search methods that navigate the generative manifold for high-quality samples. We formalize sample evaluation using a verifier function $f(\boldsymbol{x}_0)$ defined on *ground truth* samples, which measures the quality of the sample. Such verifiers could be reward functions (Xu et al., 2023a), Q-functions (Lu et al., 2023), classifier conditions $p(c|\boldsymbol{x}_0)$ (Ye et al., 2024; Dhariwal & Nichol, 2021), and VLMs (Huang et al., 2023).

To efficiently search the generative space of diffusion models, we revisit classical search principles. To capture diverse modes in the complex distributions generated by diffusion models, we view sampling as traversing a search tree, employing BFS and DFS to progressively explore states during denoising. Similar to best-first search, we evaluate intermediate states $\boldsymbol{x}_t$ with the verifier $f(\boldsymbol{x}_{0|t})$, prioritizing high-quality paths and efficiently allocating compute via branching and backtracking.

---

*Corresponding author. Contact ydu@seas.harvard.edu

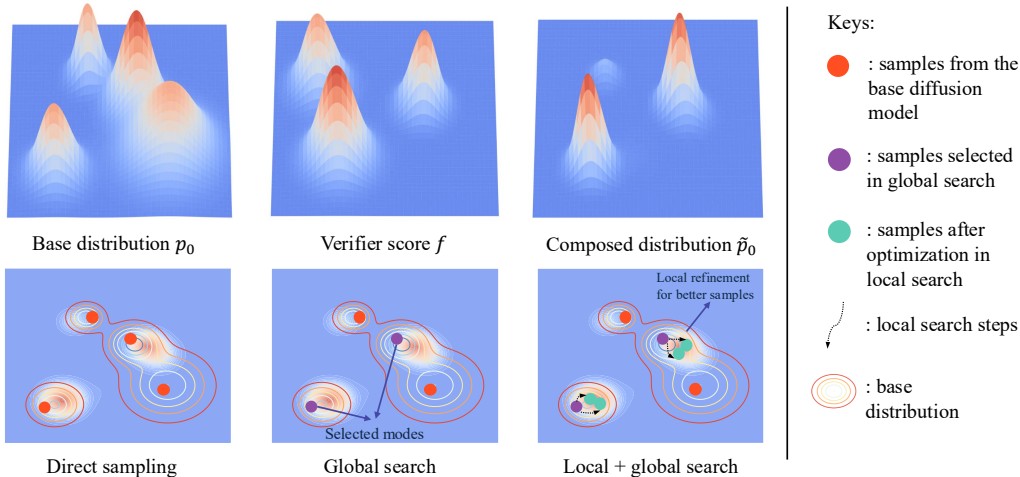

Figure 1: **Illustration of our search framework**. **Bottom left**: direct sampling results in samples with low verifier scores. **Bottom middle**: global search identifies high score modes within the base distribution. **Bottom right**: local search further optimizes the samples for higher quality, driven by the gradient signal.

To go refine the base distribution and obtain higher-quality samples, we perform local search via Langevin MCMC, exploring the neighborhood of current samples under guidance from both the verifier gradient and the diffusion model's "score function" (Vincent, 2011). By jointly optimizing the compositional objective of the diffusion model and the verifier (Du et al., 2023), our local search surpasses the capabilities of the base model. An overview of our framework is shown in Fig. 1.

Recent works scale diffusion model inference via particle-based SMC (Kim et al., 2025; Singhal et al., 2025a; Wu et al., 2023a; Li et al., 2024) and tree-based methods (Guo et al., 2025; Li et al., 2025b), which explores the diffusion sampling process with a fixed schedule. We generalize the particle-based methods (Kim et al., 2025; Singhal et al., 2025a; Li et al., 2024) with a BFS-based framework, clarifying prior design choices and providing an improved BFS baseline. Inspired by DFS, we add adaptive backtracking to allocate compute adaptively, surpassing BFS in efficiency and adaptivity. The adaptive backtrack schedule also outperforms the fixed noise injection in Ma et al. (2025), highlighting the importance of adaptivity in inference scaling. While global search remains limited to base-distribution modes, scaling local search with Langevin MCMC explores high-reward regions in neighborhoods of the base model samples, proving effective in challenging decision-making tasks. Finally, we jointly scale local and global search with the classical AI search principles, demonstrating superior performance over prior methods, which scale local and global search in isolation.

Our key contributions are summarized as follows:

**i)** For global tree search, we elucidate the design space of prior BFS-style methods and provide an improved BFS design. We further present the first adaptive DFS algorithm for diffusion inference scaling, offering superior efficiency and adaptivity.

**ii)** We introduce a theoretically grounded local search method using annealed Langevin MCMC, demonstrating superior performance in challenging domains.

**iii)** We propose a unified framework for inference scaling that integrates both global and local search within the classical AI search paradigm. By jointly scaling local and global search for the first time, we advance the Pareto frontier of inference-time scaling across diverse domains.

## 2 RELATED WORKS

Here, we provide a brief overview of inference-time scaling with diffusion models. For a more comprehensive literature review and discussion of concurrent works, see Appendix B.

**Particle-based scaling methods**. Recent works (Kim et al., 2025; Singhal et al., 2025a) propose SMC-style particle filtering methods, scaling inference compute by increasing the number of particles

and improving efficiency via resampling. Kim et al. (2025) propose biasing the transition kernel using verifier gradients and incorporating both the proposal and base transition probabilities during resampling. Similarly, tree-search-based methods (Li et al., 2024; 2025b; Guo et al., 2025) evaluate intermediate nodes and select promising candidates. Both approaches can be seen as special cases of our BFS tree search framework, which denoise a set of particles in parallel, applying branching and filtering based on intermediate reward signals. SoP (Ma et al., 2025) proposes searching over sampling paths by adding M noise samples to each noisy particle and then denoising, thereby exploring neighboring sampling paths. Although this procedure modifies the denoising process, all particles still follow the same fixed denoising schedule. In addition to linear BFS-style algorithms, we also propose DFS-style algorithms with non-linear adaptive backtracking through noise injection and a score-dependent backtracking schedule, demonstrating superior performance over prior methods.

Concurrent with our work, Jain et al. (2025) additionally incorporate value backup from MCTS into the tree search for diffusion models, leveraging information from previous sampling paths, and enabling adaptive compute allocation with the design of an anytime algorithm. Lee et al. (2025) propose backtracking by sending fully denoised particles to all noise levels. The adaptive termination condition is designed based on the reward distribution of the denoised particles to ensure sufficient exploration. In contrast, our DFS determines backtrack noise level by the score of the particle, thus reducing excessive compute consumption on easy instances. In discrete diffusion models, Dang et al. (2025) introduce Particle Gibbs Sampling for inference scaling, which outperforms SMC-based approaches (Singhal et al., 2025a). However, their method is not directly applicable to our setting.

**Gradient-based guidance methods**. To sample from a conditional distribution, classifier guidance (Dhariwal & Nichol, 2021) utilizes the gradient from a trained noise-dependent classifier to compute the conditional score function (Song et al., 2020b). However, such noise-dependent classifier requires additional training and data collection. To use classifiers defined on clean samples, training-free guidance methods (Ye et al., 2024; Chung & Ye, 2022; Song et al., 2023; Yu et al., 2023; He et al., 2023) approximates noisy conditional probability using the denoised output $\boldsymbol{x}_{0|t}$. Such methods are inheritability biased due to their first order approximation. Different from prior methods that rely on the conditional diffusion process, we sample from the compositional distribution via annealed Langevin MCMC, which provides asymptotically exact sampling without any training.

## 3 BACKGROUNDS

### 3.1 DIFFUSION PROBABILISTIC MODELS

Suppose we have $D$-dimensional random variable $\boldsymbol{x}_0 \in \mathbb{R}^D$ with distribution $p_0(\boldsymbol{x}_0)$. Diffusion models (Ho et al., 2020; Song et al., 2020a) and the more general flow models (Lipman et al., 2022; Albergo & Vanden-Eijnden, 2022) are generative models that turn noise into data via a stochastic process $\{\boldsymbol{x}_t\}_{t=0}^T$. The forward "noising" process with $t > s$ can be defined as:

$$q(\boldsymbol{x}_t|\boldsymbol{x}_s) = \mathcal{N}\left(\boldsymbol{x}; \frac{\alpha_t}{\alpha_s}\boldsymbol{x}_s, \alpha_t^2\left(\frac{\sigma_t^2}{\alpha_t^2} - \frac{\sigma_s^2}{\alpha_s^2}\right)\boldsymbol{I}\right). \tag{1}$$

where $\alpha_t, \sigma_t$ are referred as the noise schedule with $\alpha_0 = \sigma_T = 1, \alpha_T = \sigma_0 = 0$. We can thus write the random variables $\boldsymbol{x}_t$ as an interpolation between data and noise (Ma et al., 2024):

$$\boldsymbol{x}_t = \alpha_t \boldsymbol{x}_0 + \sigma_t \boldsymbol{\epsilon},$$

and denote $q_t(\boldsymbol{x}_t)$ as the marginal distribution of $\boldsymbol{x}_t$. To model the reverse "denoising" process, we train the model using the denoising objective (Ho et al., 2020):

$$L(\theta) = \mathbb{E}_{t,\boldsymbol{x}_0,\boldsymbol{\epsilon}}\left[\boldsymbol{\epsilon}_\theta(\boldsymbol{x}_t, t) - \boldsymbol{\epsilon}\right],$$

which is equivalent of learning the score function of $q_t(\boldsymbol{x}_t)$ (Vincent, 2011), as the ground truth of $\boldsymbol{\epsilon}_\theta(\boldsymbol{x}_t, t)$ is $-\sigma_t \nabla_{\boldsymbol{x}_t} \log q_t(\boldsymbol{x}_t)$. To generate samples, we transform noise into data via the reverse transition kernel $p_\theta(\boldsymbol{x}_{t-1}|\boldsymbol{x}_t)$. In practice, we either sample $\boldsymbol{x}_{t-1}$ using deterministic samplers like DDIM (Song et al., 2020a):

$$\boldsymbol{x}_{t-1} = \frac{\alpha_{t-1}}{\alpha_t}(\boldsymbol{x}_t - \sigma_t \boldsymbol{\epsilon}_\theta(\boldsymbol{x}_t, t)) + \sigma_{t-1}\boldsymbol{\epsilon}_\theta(\boldsymbol{x}_t, t)$$

or stochastic samplers like DDPM (Ho et al., 2020; Nichol & Dhariwal, 2021):

$$p_\theta(\boldsymbol{x}_{t-1}|\boldsymbol{x}_t) = \mathcal{N}(\boldsymbol{x}_{t-1}; \boldsymbol{\mu}_\theta(\boldsymbol{x}_t, t), \boldsymbol{\Sigma}_\theta(\boldsymbol{x}_t, t)).$$

## 3.2 Compositional and Controllable Generation of DPMs

Given a base diffusion model with data distribution $p_0(\boldsymbol{x}_0)$, one may wish to sample $\boldsymbol{x}_0$ with some constraints or conditions $f(\boldsymbol{x}_0)$. Exact diffusion sampling from the composed distribution $\tilde{p}_0(\boldsymbol{x}_0) \propto p_0(\boldsymbol{x}_0)f(\boldsymbol{x}_0)$ would require training a time-dependent $f$ on data generated by $p_0$ (Dhariwal & Nichol, 2021; Lu et al., 2023), which may not be applicable in practice. Thus, we adopt optimization based methods to approximate the target distribution.

**Compositional generation via annealed Langevin MCMC**. When sampling from a compositional distribution composed of multiple probability distributions, $\tilde{p}_0(\boldsymbol{x}_0) \propto p_0(\boldsymbol{x}_0)\hat{p}_0(\boldsymbol{x}_0)$, Du et al. (2023) proposes annealed Langevin MCMC sampling. In this approach, a sequence of annealed distributions $\tilde{q}_t(\boldsymbol{x}_t) \propto q_t(\boldsymbol{x}_t)\hat{q}_t(\boldsymbol{x}_t)$ is constructed with $\hat{q}_t$ corresponds to the diffusion process of $\hat{p}_0$, and samples are drawn using Langevin dynamics (Welling & Teh, 2011):

$$\boldsymbol{x}_t^{i+1} = \boldsymbol{x}_t^i + \eta \nabla_{\boldsymbol{x}} \log \tilde{q}_t(\boldsymbol{x}_t^i) + \sqrt{2\eta}\boldsymbol{\epsilon}^i, \quad \boldsymbol{\epsilon}^i \sim \mathcal{N}(\boldsymbol{0}, \boldsymbol{I}). \tag{2}$$

Since the distribution of $\boldsymbol{x}_t^i$ converges to $\tilde{q}_t(\boldsymbol{x}_t)$ asymptotically as $i \to \infty$, $\eta \to 0$, we can sample from $\tilde{p}_0(\boldsymbol{x}_0)$ following the annealing path $\{\tilde{q}_t\}_{t=0}^T$ with $\tilde{q}_0 = \tilde{p}_0$. Moreover, since the score of $\tilde{q}_t$ can be directly computed by composing the score of two distributions:

$$\nabla_{\boldsymbol{x}} \log \tilde{q}_t(\boldsymbol{x}_t) = \nabla_{\boldsymbol{x}} \log q_t(\boldsymbol{x}_t) + \nabla_{\boldsymbol{x}_t} \log \hat{q}_t(\boldsymbol{x}_t),$$

thus do not require extra training. We point out that the product distribution $\tilde{q}_t(\boldsymbol{x}_t)$ does not represent a valid diffusion process, thus inverse diffusion sampling via $\nabla_{\boldsymbol{x}} \log \tilde{q}_t$ is impossible.

**Controllable generation through training-free guidance**. During the sampling process, training-free guidance propose to update $\boldsymbol{x}_t$ using gradient ascent

$$\tilde{\boldsymbol{x}}_t = \boldsymbol{x}_t + \boldsymbol{\Delta}_t, \quad \boldsymbol{\Delta}_t = \rho_t \nabla_{\boldsymbol{x}_t} \log f(\boldsymbol{x}_{0|t}) + \mu_t \alpha_t \nabla_{\boldsymbol{x}_{0|t}} \log f(\boldsymbol{x}_{0|t}). \tag{3}$$

where $\boldsymbol{x}_{0|t} = \mathbb{E}[\boldsymbol{x}_0|\boldsymbol{x}_t] = \frac{\boldsymbol{x}_t - \sigma_t \boldsymbol{\epsilon}_\theta(\boldsymbol{x}_t, t)}{\alpha_t}$. This method approximates the intractable posterior with the posterior mean: $\mathbb{E}_{\boldsymbol{x}_0|\boldsymbol{x}_t}[f(\boldsymbol{x}_0)] \approx f(\mathbb{E}[\boldsymbol{x}_0|\boldsymbol{x}_t])$. To enhance the guidance strength, Yu et al. (2023) propose to apply a recurrence strategy, which first samples $\boldsymbol{x}_{t-1}$ via $p_\theta(\boldsymbol{x}_{t-1}|\boldsymbol{x}_t)$, add the guidance gradient, then add noise back to $\boldsymbol{x}_t$ through the forward process $q_t(\boldsymbol{x}_t|\boldsymbol{x}_{t-1})$:

$$\boldsymbol{x}_{t-1}^i \sim p_\theta(\cdot|\boldsymbol{x}_t^i), \quad \tilde{\boldsymbol{x}}_{t-1}^i = \boldsymbol{x}_{t-1}^i + \frac{\alpha_{t-1}}{\alpha_t}\boldsymbol{\Delta}_t, \quad \boldsymbol{x}_t^{i+1} \sim q_t(\cdot|\tilde{\boldsymbol{x}}_{t-1}^i), \quad i = 1, 2, \cdots, N_{\text{recur}}, \tag{4}$$

with $N_{\text{recur}}$ being the total number of recurrence steps.

## 4 Methods

**Problem Formulation**. Given a pretrained diffusion model $\boldsymbol{\epsilon}_\theta(\boldsymbol{x}_t, t)$ with a base distribution $p_0(\boldsymbol{x}_0)$, at test-time, we often wish to optimize the generation process to satisfy task-specific objectives. For example, RL may require generating high-value actions, image synthesis may seek constraint-satisfying images, and trajectory generation may demand physically valid outputs. In this paper, we are interested in how to scale test-time inference to follow such objectives.

We consider an inference-time scaling strategy that adjusts the sampling process based on a verifier function. Specifically, we define a verifier $f(\boldsymbol{x}_0) : \mathbb{R}^D \to \mathbb{R}^+$ which specifies the degree to which samples optimize a specified objective. We then aim to bias sampling toward regions of the sample space where $f(\boldsymbol{x}_0)$ is high. This leads to the objective of sampling from a compositional distribution that combines the original model distribution with the verifier:

$$\tilde{p}_0(\boldsymbol{x}_0) \propto p_0(\boldsymbol{x}_0)f(\boldsymbol{x}_0)^\lambda, \tag{5}$$

where $\lambda$ controls the weight of verifier scores.

Since exact sampling from the distribution is often impractical, we aim to search the manifold for the target samples at *inference time*, both *globally* and *locally*. First, we explore the diverse modes in the complex generative landscape of diffusion models through global graph search algorithms. However, global search alone can not generate samples beyond the pretrained model. We then propose to search the vicinity of the sample using hill-climbing style local search methods, guided by the verifier gradient. By combining local search with global search, we can avoid being stuck in local optima, advancing the performance of the base model by sampling beyond the pretrained modes.

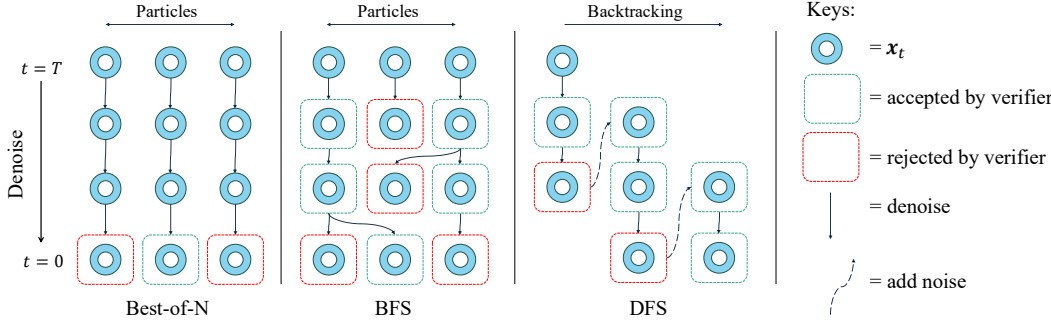

Figure 2: **Illustration of global tree search algorithms.**

## 4.1 GLOBAL SEARCH FOR MODE IDENTIFICATION

To efficiently explore the modes of a diffusion model, we represent the Markov chain of its denoising process as a *fixed-depth tree*, where the transition kernel $\tilde{p}_\theta(\boldsymbol{x}_{t-1}|\boldsymbol{x}_t)$ may correspond to either deterministic or stochastic samplers. This abstraction enables the use of classical tree-search heuristics to design compute-efficient exploration methods.

A simple baseline is best-of-$N$ (BoN) sampling: generate $N$ independent trajectories and select the one with the highest verifier score at the final step. While straightforward, this approach discards valuable information from intermediate stages. To improve efficiency, inspired by best-first search (Pearl, 1984), we evaluate intermediate particles using the denoised estimate $\boldsymbol{x}_{0|t}$. This allows dynamic allocation of computation based on particle quality, using techniques such as branching and backtracking. We next describe two representative efficient search strategies—breadth-first search (BFS) and depth-first search (DFS)—adapted for diffusion model sampling. By prioritizing the expansion of nodes with higher score estimates and backtracking from low-quality regions of the tree, we can more effectively navigate the generative space and identify high-quality modes. An illustration of the tree search methods is provided in Fig. 2.

### 4.1.1 UNIFIED BFS-STYLE LINEAR SEARCH

Inspired by breadth-first search (BFS), which expands nodes level by level, we denoise a set of particles in parallel at each noise level. We score each intermediate particle $\left\{\boldsymbol{x}_t^k\right\}_{k=1}^N$ using estimates of its verifier score $f(\boldsymbol{x}_{0|t}^k)$, and dynamically reallocate computation by sampling more children $n_t^k$ for high-scoring nodes. We provide a general design space for schedules, scoring functions, and resampling strategies that unifies previous tree-search-based and particle-based baselines such as SVDD (Li et al., 2024), DAS (Kim et al., 2025), and FK-steering (Singhal et al., 2025b). The pseudocode is shown in Alg. 3, and more details can be found in Sec. D.1.

**Tempering**. To mitigate score estimation bias in early steps, Kim et al. (2025) increases weights on smaller time steps with $\tau_T < \tau_{T-1} \cdots < \tau_0$, re-weighting scores with $\tau_t f(\boldsymbol{x}_{0|t}^k)$. Li et al. (2024) samples only from the top-scoring particle, i.e., $\tau_t = \infty$. We consider three different tempering design choices: **Constant** : $\tau_t = \tau$, **Increase** : $\tau_t = \left((1+\gamma)^{T-t} - 1\right)\tau$, **Inf** : $\tau_t = \infty$ .

**Scoring**. Following Kim et al. (2025); Singhal et al. (2025b), we propose to evaluate intermediate particles $\widehat{f}(\boldsymbol{x}_t^k)$ using functions of the estimated rewards $f(\boldsymbol{x}_{0|s}^k)$ along the sampling path $s \in [t, T]$. This approach accounts for reward variation during sampling by considering not only the current reward but also its trajectory. Specifically, we define the following scoring functions $\widehat{f}(\boldsymbol{x}_t^k)$ based on $f(\boldsymbol{x}_{0|s}^k)$ for $s \geq t$: **Current** : $\tau_t f(\boldsymbol{x}_{0|t}^k)$, **Difference** : $\tau_t f(\boldsymbol{x}_{0|t}^k) - \tau_{t+1} f(\boldsymbol{x}_{0|t+1}^k)$, **Max** : $\max_{s \geq t} \tau_s f(\boldsymbol{x}_{0|s}^k)$.

**Resampling**. Given $\widehat{f}(\boldsymbol{x}_t^k)$, we allocate particles as $\left(n_t^1, \cdots, n_t^N\right) = \text{Resample}\left(N; w_t^1, \cdots, w_t^N\right)$, where $n_t^k$ is the number of children for $\boldsymbol{x}_t^k$ and $w_t^k = \text{softmax}\,\widehat{f}(\boldsymbol{x}_t^k)$ is the softmax score of particle $k$. **Multinomial** resampling (Singhal et al., 2025b) samples $n_t^k$ independently from the multinomial distribution $w_t^k$, but suffers from large variance. Kim et al. (2025) adopts the Srinivasan sampling process (SSP) resampling (Alg. 4) for reduced variance. We compare the baseline **Multinomial** resampling and the variance-reduced **SSP** (Kim et al., 2025); see Gerber et al. (2020) for other methods.

Prior methods can be seen as special cases of BFS: SVDD (Li et al., 2024) = **BFS** (**Inf**, **Current**, **Multinomial**); DAS (w/o grad) (Kim et al., 2025) = **BFS** (**Increase**, **Difference**, **SSP**)[1]; FK-steering (Singhal et al., 2025a) = **BFS** (**Constant**, **Max**, **Multinomial**). Ablations (Sec. 5.1) show **SSP** resampling is key for performance, and our improved **BFS** (**Increase**, **Max**, **SSP**) consistently outperforms prior methods in global search efficiency.

### 4.1.2 DFS-STYLE NON-LINEAR SEARCH

Depth-first search (DFS) explores one branch of the search tree as deeply as possible before backtracking. In our setting, this corresponds to iteratively denoising a single particle until its verifier score drops below a predefined threshold: $f(\boldsymbol{x}_{0|t}) \leq \delta_t$, where $\delta_t$ is a threshold representing the quality requirement of the users.

Once the constraint is violated, the algorithm backtracks by reintroducing noise, moving to a higher noise level $t_{\text{next}} = t + \Delta_T$ using the forward diffusion process $q(\boldsymbol{x}_{t_{\text{next}}}|\boldsymbol{x}_t)$ in Eq. 1. This allows the model to restart the denoising process from a different region of the manifold, encouraging exploration of diverse modes. Compared with the fixed noise injection schedule used in SoP (Ma et al., 2025), DFS adopts a score-adaptive exploration strategy that enables early backtracking for global restarts and prevents excessive compute from being allocated to easy problem instances. Together, these mechanisms enable more adaptive and computationally efficient global search.

A key strength of DFS is its ability to allocate compute adaptively: difficult prompts and low-quality trajectories naturally trigger more backtracking and exploration, while easier instances are solved more directly. This dynamic behavior is driven purely by the verifier signal, without needing to know the difficulty in advance as in Snell et al. (2024). Also, the threshold acts as a control knob for users to balance output quality and computation resources, where higher threshold automatically scales compute for better output. As shown in Sec. 5.2, this adaptive strategy leads to substantial gains in efficiency and performance over prior methods, and even our improved BFS implementation. The pseudocode is in Alg. 5 and more details can be found in Sec. D.2.

## 4.2 SCALING LOCAL SEARCH VIA LANGEVIN MCMC WITH VERIFIER GRADIENT

Global search can efficiently discover the high score modes within the base diffusion model, but struggles to generate higher quality samples with refined details and from low density regions. Thus, we aim to sample from the compositional distribution $\tilde{p}_0$ in Eq. 5 for higher quality samples. To optimize the compositional objective, we conduct local-search with hill-climbing methods, aiming to find the local maximum with high $\tilde{p}_0$. Specifically, we view the sampling problem as compositional optimization in measure space (Wibisono, 2018), and follow the gradient flow of KL-divergence, performing Langevin MCMC steps (details see Appendix. C.1).

Similar to annealed Langevin MCMC in Du et al. (2023), we could construct a series of annealed functions $\hat{f}_t(\boldsymbol{x}_t)$ with $\hat{f}_0(\boldsymbol{x}_0) = f(\boldsymbol{x}_0)$. Then we sample from the distributions $\tilde{q}_t(\boldsymbol{x}_t) \propto q_t(\boldsymbol{x}_t)\hat{f}(\boldsymbol{x}_t)$ through Langevin MCMC in Eq. 2 (details see Appendix. C.2). Alternatively, training-free guidance in Eq. 3 utilizes the gradient of $f(\boldsymbol{x}_{0|t})$ to optimize $\boldsymbol{x}_t$, which can be computed directly using the diffusion model output. However, naive gradient updates have been observed to produce OOD and adversarial samples (Shen et al., 2024). In Ye et al. (2024), recurrence (Eq. 4) was found to help avoid adversarial samples in challenging guidance tasks, though its theoretical underpinnings remain poorly understood. We unify these two approaches by demonstrating that training-free guidance with recurrence, in the continuous limit, constitutes an instance of Langevin MCMC. For details see Appendix. C.3, and a rigorous convergence bound is in Theorem. 1.

**Proposition 1.** *In the continuous limit where the number of diffusion denoising steps $T \to \infty$, training-free guidance with recurrence is equivalent to running Langevin MCMC on a series of annealed distributions $\{\tilde{q}_t(\boldsymbol{x}_t)\}_{t=0}^T$, with $\tilde{q}_0(\boldsymbol{x}_0) = \tilde{p}_0(\boldsymbol{x}_0) \propto p_0(\boldsymbol{x}_0)f(\boldsymbol{x}_0)^\lambda$.*

Thus, the recurrence step (without guidance) can be interpreted as Langevin MCMC applied to the original distribution of the diffusion model $q_t(\boldsymbol{x}_t)$, and the guidance term $\boldsymbol{\Delta}_t$ in Eq. 3 then serves

---

[1]The original implementation of DAS uses a verifier–gradient–guided proposal kernel and incorporates both the proposal probability and the base model probability during resampling. For a fair comparison, we disable the gradients of all methods in Sec. 5.1, and provide additional results in Sec. E.1

as defining a practical annealing path $\hat{f}_t(\boldsymbol{x}_t)$ that bias the sampling path towards high reward areas beyond the modes of the base model. We are the *first* to propose this theoretical unification of the two lines of work, providing insights into efficient local search of diffusion models via gradients.

We implement local search by parameterizing the reverse transition kernel $\tilde{p}_\theta(\boldsymbol{x}_{t-1}|\boldsymbol{x}_t)$ as a sequence of Langevin MCMC steps (Eq. 2), followed by a denoising step using DDIM (Eq. 11) or DDPM (Eq. 12); see Appendix C.5 for details. Unlike classifier-guidance or naive training-free guidance, which apply only gradient guidance in the denoising step, our approach incorporates explicit Langevin MCMC steps. In Sec. 5.3, we scale the number of local search steps for the first time and observe substantial improvements over pretrained models across multiple tasks.

## 5 EXPERIMENTS

In this section, we apply inference-time scaling with our search strategy across a range of domains. In Sec. 5.1, we present a strengthened BFS baseline that outperforms previous particle-based methods. In Sec. 5.2, we demonstrate the adaptivity and efficiency of our DFS method. In Sec. 5.3, we scale up local search in challenging decision-making domains, highlighting the importance of jointly scaling local and global search. In Sec. 5.4, we propose a double-verifier strategy to mitigate the reward hacking problem in inference-scaling.

### 5.1 ELUCIDATING THE DESIGN SPACE OF BFS

In this section, we explore the design choices of BFS, and present an improved **BFS** (**Increase**, **Max**, **SSP**) over prior particle-based or tree-search-based methods (Singhal et al., 2025b; Kim et al., 2025; Guo et al., 2025; Li et al., 2024; 2025b), where we disable the gradient guidance in DAS (Kim et al., 2025) as in other methods. We also compare with SoP (Ma et al., 2025) which adopts noise injection. To ensure a fair comparison, we directly use the official implementation of FK-steering (Singhal et al., 2025a), with the ImageReward (Xu et al., 2023a) verifier and prompts from the ImageReward paper[2]. For ablation of design choices, we use the SD v1.5 model. For more details, see Appendix E.1.

Table 1: Ablation of BFS design choices. For each section, we vary one design choice while keeping the others fixed. **Left**: Resampling methods (with Constant tempering, Max scoring). **Center**: Scoring functions (with SSP resampling, Constant tempering). **Right**: Tempering schedules (with SSP resampling, Max scoring).

| | Resampling | | | Scoring | | | Tempering | | |
|---|---|---|---|---|---|---|---|---|---|
| N | BoN | Multinomial | SSP | Current | Difference | Max | Increase | Inf | Constant |
| 4 | $0.702 \pm 0.057$ | $0.743 \pm 0.037$ | $\mathbf{0.834 \pm 0.041}$ | $0.812 \pm 0.037$ | $0.823 \pm 0.036$ | $\mathbf{0.834 \pm 0.041}$ | $\mathbf{0.882 \pm 0.029}$ | $0.667 \pm 0.076$ | $0.834 \pm 0.041$ |
| 8 | $0.896 \pm 0.031$ | $0.926 \pm 0.042$ | $\mathbf{1.032 \pm 0.035}$ | $0.996 \pm 0.029$ | $1.013 \pm 0.032$ | $\mathbf{1.032 \pm 0.035}$ | $\mathbf{1.087 \pm 0.031}$ | $0.775 \pm 0.087$ | $1.032 \pm 0.035$ |

As mentioned in Sec. 4.1.1, we experiment with different resampling, scoring, and tempering design choices. We begin with **BFS** (**Constant**, **Max**, **Multinomial**) and evaluate various resampling strategies. As shown in Table 1 (left), **SSP** significantly outperforms naive multinomial resampling, and we adopt it in our design. We then assess scoring functions and tempering schedules under SSP resampling, as reported in Table 1 (center) and 1 (right). Here, we observe modest improvements with **Max** scoring and **Increase** tempering, leading to our final design: **BFS** (**Increase**, **Max**, **SSP**).

Table 2: Comparison of our BFS with prior methods

| Model | N | BoN | FK (Singhal et al., 2025a) | DAS (w/o grad) (Kim et al., 2025) | TreeG (Guo et al., 2025) | SVDD (Li et al., 2024) | SoP (Ma et al., 2025) | DSearch (Li et al., 2025b) | BFS (ours) |
|---|---|---|---|---|---|---|---|---|---|
| SD v1.5 | 4 | $0.702 \pm 0.057$ | $0.743 \pm 0.037$ | $0.878 \pm 0.028$ | $0.860 \pm 0.033$ | $0.667 \pm 0.076$ | $0.688 \pm 0.024$ | $0.836 \pm 0.032$ | $\mathbf{0.882 \pm 0.029}$ |
| SD v1.5 | 8 | $0.891 \pm 0.031$ | $0.926 \pm 0.042$ | $1.052 \pm 0.033$ | $1.023 \pm 0.018$ | $0.775 \pm 0.087$ | $0.884 \pm 0.022$ | $1.011 \pm 0.019$ | $\mathbf{1.087 \pm 0.031}$ |
| SD XL | 4 | $1.085 \pm 0.013$ | $1.131 \pm 0.022$ | $1.181 \pm 0.023$ | $1.152 \pm 0.023$ | $1.036 \pm 0.062$ | $1.076 \pm 0.014$ | $1.139 \pm 0.018$ | $\mathbf{1.194 \pm 0.024}$ |
| SD XL | 8 | $1.198 \pm 0.021$ | $1.251 \pm 0.011$ | $1.265 \pm 0.019$ | $1.261 \pm 0.021$ | $1.225 \pm 0.027$ | $1.185 \pm 0.012$ | $1.252 \pm 0.019$ | $\mathbf{1.291 \pm 0.018}$ |
| FLUX.1 | 4 | $1.113 \pm 0.015$ | $1.145 \pm 0.017$ | $1.194 \pm 0.013$ | $1.178 \pm 0.018$ | $1.069 \pm 0.035$ | $1.104 \pm 0.011$ | $1.169 \pm 0.022$ | $\mathbf{1.203 \pm 0.013}$ |

To compare our improved **BFS** (**Increase**, **Max**, **SSP**) with prior baselines, we additionally experiment with the SDXL model (Podell et al., 2023) and FLUX.1 dev[3], which differs from the model used in our ablations. As shown in Table 2, our improved BFS consistently outperforms previous methods across compute budgets and models. Among the baselines, DAS with out gradient guidance (Kim et al., 2025) exhibits the smallest performance gap due to a similar design space with SSP resampling, while SoP (Ma et al., 2025) underperforms in global search efficiency because it allocates compute uniformly across all particles during the noising and denoising process when searching over paths. In the following experiments, we use the improved BFS as our strengthened baseline.

---

[2]We follow the settings of Table 1 in Singhal et al. (2025a)

[3]https://huggingface.co/black-forest-labs/FLUX.1-dev

## 5.2 Adaptive and Efficient Inference-Scaling with DFS

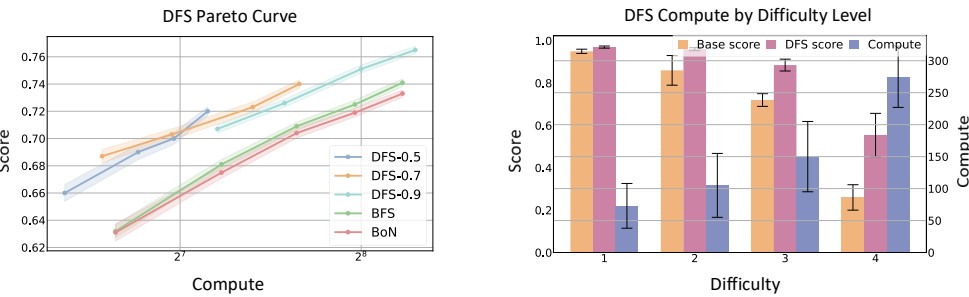

Figure 3: **CompBench text-to-image results with DFS. Left**: the Pareto curve of DFS, with DFS-$\delta$ denotes DFS with threshold $\delta_t = \delta$. **Right**: average compute allocation by DFS for prompts of increasing difficulty.

In this section, we evaluate the adaptivity and efficiency of DFS on the CompBench dataset (Huang et al., 2023), using the SSD-1B model (Gupta et al., 2024) and a VLM (Li et al., 2022) as our verifier. Through these experiments, we address the following questions:

- *Can DFS outperform Best-of-N and prior particle-based methods?* As shown in Fig. E.2 (left), DFS consistently outperforms Best-of-N and also our improved BFS design across different compute budgets, with up to $30\%$ less computational cost.

- *Can DFS adjust compute allocation with different thresholds?* We evaluate DFS across a wide range of practical threshold values ($0.5, 0.7,$ and $0.9$) and find that lower thresholds automatically allocate less compute, while higher thresholds scale up compute for better quality. DFS consistently outperforms baseline methods across all threshold choices, demonstrating the robustness of our method for different hyperparameters.

- *Can DFS dynamically adjust compute allocation for different instances?* We measure the computational cost of DFS on prompts of varying difficulty under fixed thresholds, where the difficulty of a prompt is defined as the average score of the base model over four random samples. As shown in Fig. 3 (right), harder prompts with lower scores naturally consume more compute, without requiring prior knowledge of difficulty as in Snell et al. (2024). This adaptive design reduces compute waste on easy instances and allocates additional compute to achieve greater improvement on harder problems, as evidenced by the score increase in Fig. 3 (right).

The detailed setup is provided in Appendix E.2, where we provide visualizations in Fig. 7 and comparisons with SoP (Ma et al., 2025) in Fig. 8. Unlike linear-search methods that use a fixed exploration schedule, DFS offers higher efficiency and adaptivity, which may be of independent interest to the broader community.

## 5.3 Joint Scaling Local and Global Search

Although global search methods such as BFS and DFS can efficiently explore the generative space of the diffusion model, they are restricted to the modes of the base distribution and therefore cannot exceed the capabilities of the base model. To optimize the compositional objective in Eq. 5 and sample from high-reward regions beyond the base model, we propose scaling up local search steps via annealed Langevin MCMC, introducing a new scaling dimension for diffusion models. We validate the effectiveness of scaling local search in challenging decision-making domains, such as long-horizon planning and offline RL.

**Baselines**. To evaluate the effectiveness of scaling local search steps, we compare our method with particle-based DAS (Kim et al., 2025), which also leverages verifier gradients but only incorporates gradient guidance without recurrent local search. We also compare against the state-of-the-art training-free guidance method TFG (Ye et al., 2024), which performs multiple recurrence steps but lacks any global search. For fairness, compute is measured as the total NFEs of both local and global search. As shown in the following experiments, scaling local and global search separately yields suboptimal performance, whereas our joint scaling strategy establishes a new Pareto frontier.

### 5.3.1 LONG HORIZON PLANNING

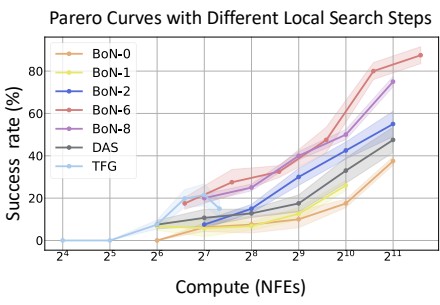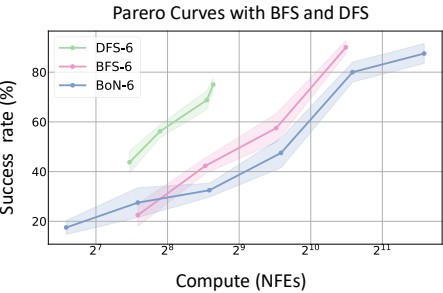

Figure 4: **Pareto curves of local search. Left**: Pareto curves of best-of-N with different local search steps, where BoN-$i$ denotes $i$ local search steps. **Right**: Pareto curves of BFS and DFS with 6 local search steps.

Diffusion models have been widely adopted in planning for trajectory synthesis (Ubukata et al., 2024). We evaluate long-horizon planning in a challenging PointMaze environment, using the base model trained following Diffuser (Janner et al., 2022), with the verifier defined as the total number of collisions between the trajectory and maze walls (see Appendix E.3 for details). (Luo et al., 2024; Marcucci et al., 2023). As shown in Fig. 5 (left), naive sampling without local search often resulted in failed trajectories that violates the maze layout in some of the segments. When scaling up local search steps as in Fig. 5 (right), we are able to generate successful plans free of violation.

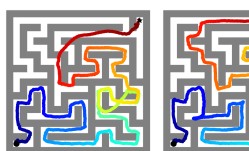

Figure 5: Illustration of Maze layout and task, with failed trajectory (left) and successful sample (right).

To evaluate the compute efficiency of recurrent steps, we observe in Fig. 4 (left) that scaling up local search with BoN improves the overall Pareto frontier and significantly outperforms baseline methods. Scaling local search only as in TFG (Ye et al., 2024) is efficient with a low compute budget but fails to scale with increased compute, as local search alone can become trapped in local optima. DAS (Kim et al., 2025) is more efficient than the corresponding BoN-0 baseline without local search, but underperforms best-of-N when more local search steps are used. In Fig. 4 (right), we show that local search can be combined with global search techniques such as BFS and DFS to further improve scaling efficiency, demonstrating the flexibility of our framework.

### 5.3.2 OFFLINE REINFORCEMENT LEARNING

Table 3: Performance on D4RL locomotion tasks, highlighting results within 5% of the maximum.

| Dataset | Environment | IQL | SfBC | DD | Diffuser | D-QL | QGPO | TFG | DAS | TTS(ours) |
|---|---|---|---|---|---|---|---|---|---|---|
| Medium-Expert | HalfCheetah | 86.7 | **92.6** | 90.6 | 79.8 | **96.1** | 93.5 | $90.2 \pm 0.2$ | **93.3 $\pm$ 0.3** | $93.9 \pm 0.3$ |
| Medium-Expert | Hopper | 91.5 | 108.6 | **111.8** | 107.2 | 110.7 | **108.0** | $100.2 \pm 3.5$ | $105.4 \pm 5.1$ | $104.4 \pm 3.1$ |
| Medium-Expert | Walker2d | **109.6** | **109.8** | 108.8 | 108.4 | 109.7 | 110.7 | $108.1 \pm 0.1$ | **111.4 $\pm$ 0.1** | **111.4 $\pm$ 0.1** |
| Medium | HalfCheetah | 47.4 | 45.9 | 49.1 | 44.2 | 50.6 | **54.1** | $53.1 \pm 0.1$ | $53.4 \pm 0.1$ | **54.8 $\pm$ 0.1** |
| Medium | Hopper | 66.3 | 57.1 | 79.3 | 58.5 | 82.4 | **98.0** | $96.2 \pm 0.5$ | $71.3 \pm 2.7$ | **99.5 $\pm$ 1.7** |
| Medium | Walker2d | 78.3 | 77.9 | **82.5** | 79.7 | 85.1 | **86.0** | $83.2 \pm 1.4$ | $83.9 \pm 0.9$ | **86.5 $\pm$ 0.2** |
| Medium-Replay | HalfCheetah | 44.2 | 37.1 | 39.3 | 42.2 | 47.5 | **47.6** | $45.0 \pm 0.3$ | $42.2 \pm 0.1$ | **47.8 $\pm$ 0.4** |
| Medium-Replay | Hopper | 94.7 | 86.2 | **100.0** | 96.8 | **100.7** | 96.9 | $93.1 \pm 0.1$ | **96.7 $\pm$ 3.0** | **97.4 $\pm$ 4.0** |
| Medium-Replay | Walker2d | 73.9 | 65.1 | 75.0 | 61.2 | **94.3** | 84.4 | $69.8 \pm 4.0$ | $63.8 \pm 2.0$ | $79.3 \pm 9.7$ |
| **Average (Locomotion)** | | 76.9 | 75.6 | 81.8 | 75.3 | **86.3** | **86.6** | 82.1 | 80.2 | **86.1** |

Recently, diffusion models have emerged as a powerful action prior in robotics due to their ability to model complex and multimodal distributions (Chi et al., 2023). However, these diffusion policies are typically trained on offline datasets and struggle to adapt to reinforcement learning or test-time requirements. Following Peters et al. (2010), we formulate the offline RL problem as sampling from a Q-regularized distribution: $\pi^*(\boldsymbol{a}|\boldsymbol{s}) \propto \mu(\boldsymbol{a}|\boldsymbol{s})e^{\beta Q_\psi(\boldsymbol{s},\boldsymbol{a})}$, where $Q_\psi$ is a learned Q-function representing preferences over actions, and $\mu$ is the behavior policy, which we model using a diffusion prior. We approach this problem from the inference-scaling perspective, composing off-the-shelf pretrained diffusion policy with ground-truth Q-functions without training.

Among the baselines, Diffuser (Janner et al., 2022), QGPO (Lu et al., 2023), and D-QL (Wang et al., 2022) require additional joint training of the diffusion model and Q-function. To demonstrate the effectiveness of our method *test-time search* (**TTS**), we allow TFG (Ye et al., 2024) and DAS (Kim et al., 2025) to use up to *twice* the compute of TTS. Since we use a small number of particles, we simply adopt BoN for global search of TTS and searched for the number of local search steps, with detailed hyper-parameters in Table 10. As shown in Table 3, TTS achieves performance comparable to training-based baselines, while DAS and TFG struggles on the Medium and Medium-Replay datasets where the model's capabilities are limited. Details see Sec. E.4.

**Advancing Online Learning with Policy Distillation**. To enhance the capabilities of the base model, we fine-tune it on actions generated by TTS and subsequently evaluate its one-shot sampling performance. We adapt the Medium datasets by replacing the original actions with those produced by TTS (see Sec. E.4.1 for details). Perhaps surprisingly, this simple offline distillation outperforms SOTA diffusion-based online RL methods such as DPPO (Ren et al., 2024),

Table 4: Results for policy distillation with TTS

| Environment | TTS-distill | DPPO |
|---|---|---|
| Halfcheetah | **51.6 ± 0.7** | 47.8 |
| Hopper | **98.8 ± 0.8** | 92.8 |
| Walker2d | **86.3 ± 0.5** | 82.7 |

as shown in Table 4. Compared with other offline RL methods such as PA-RL (Mark et al., 2024), which also finetune the policy using optimized actions, our approach relies solely on pretrained verifiers and does not require iterative joint updates of the Q-function and policy. This enables more flexible composition of pretrained foundation models for verification and further highlights the effectiveness of inference scaling compared with naive action optimization. To the best of our knowledge, this is the first work to demonstrate self-improvement via inference scaling for diffusion models using pretrained verifiers, a direction that may be of independent interest.

## 5.4 MITIGATING REWARD HACKING WITH DOUBLE VERIFIER

In this section, we show that reward hacking and over-optimization can be mitigated by a double-verifier strategy: employing separate verifiers for local and global search. As observed in Shen et al. (2024), training-free guidance with verifier gradients is vulnerable to adversarial reward hacking: gradient guidance often over-optimize the verifier, causing it to classify them as belonging to the target class despite being out-of-distribution (OOD). Inspired by double-Q learning in reinforcement learning (Van Hasselt et al., 2016), we propose a *double-verifier* approach, assigning distinct verifiers to local and global search to efficiently reduce overestimation.

Table 5: Results for ImageNet conditional generation with *double-verifier* using BoN and BFS.

| | BoN-single | | | BoN-double | | | BFS-single | | | BFS-double | | |
|---|---|---|---|---|---|---|---|---|---|---|---|---|
| N | FID↓ | Acc↑ | MSP↑ | FID↓ | Acc↑ | MSP↑ | FID↓ | Acc↑ | MSP↑ | FID↓ | Acc↑ | MSP↑ |
| 4 | 171.5 | 31.8% | 0.161 | **151.2** | **37.5%** | **0.164** | 156.2 | 36.1% | 0.169 | **145.5** | **44.3%** | **0.181** |
| 8 | 155.7 | 35.8% | 0.165 | **127.8** | **49.2%** | **0.184** | 133.3 | 46.5% | 0.183 | **118.2** | **55.9%** | **0.214** |

We evaluate the proposed double-verifier on the challenging conditional ImageNet generation task, generating target-class samples from an unconditional model guided by a pretrained classifier defined only on clean samples. We report the FID and class accuracy of 256 samples. We also report the MSP score (Hendrycks & Gimpel, 2016), with higher MSP score indicating less OOD samples. For double-verifier, we use two distinct ImageNet classifiers for local[4] and global search[5] respectively. We use another classifier[6] for evaluating class accuracy to avoid reward hacking. As shown in Table 5, using double-verifier significantly improves performance over single verifier with 2x less compute. Also, double-verifier significantly reduces OOD samples indicated by the higher MSP score, improving robustness alongside efficiency. Visualizations and details see Appendix E.5.

## 6 LIMITATIONS AND CONCLUSION

In this work, we present a unified and principled framework for inference-time scaling of diffusion models, jointly scaling local and global search. A potential limitation of our approach is that inference-time scaling still requires additional hyperparameters. To address this, heuristics such as evolutionary search can be adopted during hyperparameter tuning, which we leave for future work.

---

[4] https://huggingface.co/google/vit-base-patch16-224
[5] https://huggingface.co/google/vit-base-patch16-384
[6] https://huggingface.co/facebook/deit-small-patch16-224

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

## A    APPENDIX OVERVIEW

In Sec. B, we provide a in-depth review of literature related to inference-time scaling and diffusion models. In Sec. C, we elaborate on local search with Langevin MCMC, and in Sec. D we provide the pseudo code and design of global search algorithms BFS and DFS. In Sec. E, we provide the details of all the experiments.

## B    ADDITIONAL RELATED WORKS

### B.1    DISCUSSIONS ON CONCURRENT WORKS

Adaptive Bi-directional Cyclic Diffusion (ABCD) (Lee et al., 2025) introduces a search-based inference-scaling framework that combines elements of both DFS and BFS. Unlike DFS, which relies on a quality threshold to determine backtracking, ABCD maintains a set of particles and backtracks by propagating them across all noise levels. The termination criterion is based on whether backtracking to higher noise levels improves sample quality. Compared with DFS, ABCD achieves smaller score estimation errors since particles are evaluated after full denoising, and it explores the generative space more thoroughly by combining BFS and DFS. However, ABCD cannot adaptively allocate compute across different instances due to its fixed termination condition, and it expends unnecessary compute on easier instances because it denoises a full set of particles regardless of sample quality.

EvoSearch (He et al., 2025) proposes evolutionary search to scale inference compute for image and video generation. At selected timesteps, particles are fully denoised; high-scoring particles are retained, while low-scoring ones are perturbed via noise injection. This method improves over the FK-steering baseline (Singhal et al., 2025a).

In contrast, our work develops a uniform and principled framework grounded in classical search heuristics. BFS and DFS serve as fundamental building blocks for more advanced search methods, and, crucially, we are the first to scale local search using annealed Langevin MCMC. This local search can be seamlessly combined with global search strategies, a joint scaling approach that we show is critical for success in challenging decision-making domains.

### B.2    INFERENCE-TIME SCALING WITH CLASSICAL SEARCH

We now review the use of classical search for inference-time scaling, including applications beyond diffusion models.

Inference-time compute scaling, often viewed as "slow thinking," has long-standing roots in cognitive science, where it corresponds to "system 2" reasoning (Kahneman, 2011; Sloman, 1996). In early work, Newell et al. (1959; 1972) formalized problem solving as tree search in combinatorial spaces. More recently, Yao et al. (2023) proposed tree-of-thoughts reasoning for LLMs, enabling exploration along multiple reasoning paths using BFS and DFS as strategic search primitives.

### B.3    INFERENCE SCALING IN DIFFUSION MODELS WITHOUT DIFFERENTIABLE OPTIMIZATION

We next provide a more detailed overview of gradient-free inference-scaling approaches in diffusion models.

Inference-time compute for diffusion models is largely determined by the number of denoising steps. Xu et al. (2023b) proposed recursive restart sampling to mitigate cumulative errors, effectively scaling the number of denoising steps. More recently, Singhal et al. (2025a); Kim et al. (2025) introduced Sequential Monte Carlo (SMC) approaches (Wu et al., 2023a; Zhao et al., 2024), which can be interpreted as BFS-style algorithms from our tree-sampling perspective, since they search over all nodes at the same depth before denoising further. Notably, SMC style methods propose to use a different proposal transition kernel such as the gradient guided sampler $\mathcal{N}\left(\mu_\theta(\boldsymbol{x}_t) + \nabla_{\boldsymbol{x}_t} \log f(\boldsymbol{x}_{0|t}), \sigma_t\right)$, which we disable during our global search experiments. Their implementations differ mainly in the choice of scoring functions and resampling strategies, with Kim et al. (2025) proposing an increasing tempering schedule. Li et al. (2025b); Guo et al. (2025) proposed beam-search style tree search methods, where Li et al. (2025b) proposed to adopt dynamic beam-size and tree-width during the sampling process. Ma et al. (2025) further explored inference-time scaling for image generation

via a combination of local zero-order search and global path search, experimenting with different verifiers (oracle, self-supervised, and task-aligned) and analyzing verifier-task alignment. However, they did not systematically evaluate and improve the compute efficiency of search methods, while we demonstrate significantly improved efficiency with our novel design in search methods. Additionally, Yoon et al. (2025) proposed a Monte Carlo Tree Search-based diffusion framework tailored for trajectory synthesis, which denoises different trajectory segments at distinct steps. More recently, Jain et al. (2025) proposed to integrate the value backup in MCTS into the tree sampling and search of diffusion models, which can utilize information from low noise states and previous sampling paths. For diffusion language models, Dang et al. (2025) proposed particle Gibbs sampling methods that outperform naive SMC (Singhal et al., 2025a) approaches.

Beyond search-based methods, Li et al. (2025a) leverage the in-context learning abilities of foundation models to enable revision during sampling. Their approach exploits multimodal VLM capabilities to provide feedback on intermediate generations and trains the model to condition on both prior samples and corresponding feedback.

### B.4 GRADIENT GUIDANCE OF DIFFUSION MODELS

We review guidance methods for diffusion models that leverage gradients from verifiers or classifiers.

To sample from a conditional distribution

$$\tilde{p}_0(\boldsymbol{x}_0) \propto p_0(\boldsymbol{x}_0)\, p(c|\boldsymbol{x}_0),$$

where $p$ denotes a classifier, one can define a conditional diffusion process $\{\tilde{q}_t(\boldsymbol{x}_t)\}_{t=0}^T$ with initial distribution $\tilde{p}_0$. The corresponding conditional score function is

$$\nabla_{\boldsymbol{x}_t} \log \tilde{q}_t(\boldsymbol{x}_t) = \nabla_{\boldsymbol{x}_t} \log q_t(\boldsymbol{x}_t) + \nabla_{\boldsymbol{x}_t} \log p_t(c|\boldsymbol{x}_t),$$

where $p_t(c|\boldsymbol{x}_t) = \mathbb{E}_{\boldsymbol{x}_0|\boldsymbol{x}_t}[p(c|\boldsymbol{x}_0)]$. Classifier guidance (Dhariwal & Nichol, 2021) trains a noise-dependent classifier $p_t(c|\boldsymbol{x}_t)$ on noisy samples, using data from the diffusion model together with the base classifier. However, this training procedure is computationally expensive and impractical when the base diffusion model cannot generate target-class samples.

Training-free guidance methods (Ye et al., 2024; Chung & Ye, 2022; Song et al., 2023; Yu et al., 2023; He et al., 2023) approximate the intractable $p_t(c|\boldsymbol{x}_t)$ with $p(c|\boldsymbol{x}_{0|t}) = p(c|\mathbb{E}_{\boldsymbol{x}_0|\boldsymbol{x}_t}[\boldsymbol{x}_0])$, which can be interpreted as a first-order approximation. These methods introduce bias, though recurrence has been shown to mitigate out-of-distribution artifacts and improve performance in classifier-guided generation (Ye et al., 2024). Nevertheless, the theoretical foundations and scaling behavior of recurrent guidance remain largely unexplored.

In this work, we instead adopt Langevin MCMC sampling in place of score-based ODE sampling. This approach guarantees asymptotic exactness using only the base verifier, without requiring additional training. Relatedly, Du et al. (2024) proposed annealed energy-based models (EBMs) combined with iterative MCMC for inference scaling and reasoning, but their method cannot be applied to pretrained diffusion models and verifiers. For the first time, we establish a connection between annealed Langevin MCMC and recurrent guidance, thereby enabling inference scaling with pretrained foundation models.

## C DETAILS ABOUT LOCAL SEARCH WITH LANGEVIN MCMC

In this section we provide a comprehensive and detailed overview of (annealed) Langevin MCMC based methods used in local search, as well as proving Proposition. 1.

### C.1 LANGEVIN MCMC AS GRADIENT FLOW IN MEASURE SPACE

Following Wibisono (2018), the Langevin SDE in sample space corresponds to gradient flow of the KL-divergence in measure space. Here we provide a brief overview.

Define our target distribution that we wish to sample from as $\nu$, and the distribution of our current sample as $\rho$. We define the KL-divergence (relative entropy) as:

$$H_\nu(\rho) = \int \rho \log \frac{\rho}{\nu}\,. \tag{6}$$

Thus, sampling from $\nu$ can be seen as minimizing $H$, since the minimum of $H$ is achieved at $\rho = \nu$ with $H_\nu(\rho) = 0$. Furthermore, $\nu$ is the only stationary point of $H$ even for multimodal distributions. Thus we can sample from $\nu$ when optimizing $H$ via gradient based methods.

We have the gradient flow of $H$ in Eq. 6 follows the following PDE:

$$\frac{\partial \rho}{\partial t} = \nabla \cdot (\rho \nabla(-\log \nu)) + \Delta \rho \,, \tag{7}$$

which is known as the Fokker-Planck equation. Here, $\rho = \rho(\boldsymbol{x}, t)$ is a smooth positive density evolving through time, driven by the dynamics of the sample $\boldsymbol{x}$. The dynamics in sample space corresponding to Eq. 7 is the Langevin SDE:

$$d\boldsymbol{x}_t = \nabla \log \nu(\boldsymbol{x}_t)dt + \sqrt{2}d\boldsymbol{w}_t \,. \tag{8}$$

where $(\boldsymbol{x}_t)_{t\geq 0}$ is a stochastic process with measure $\rho_t$, and $(\boldsymbol{w}_t)_{t\geq 0}$ is standard Brownian motion. That is, if $\boldsymbol{x}_t \sim \rho_t$ evolves according to the dynamics in Eq. 8, then the measure $\rho(\boldsymbol{x}, t) = \rho_t$ evolves according to the PDE in Eq. 7, conducting gradient optimization in measure space.

In practice, we implement Eq. 8 through discretization, which is known as the unadjusted Langevin algorithm (ULA):

$$\boldsymbol{x}^{i+1} = \boldsymbol{x}^i + \eta \nabla_{\boldsymbol{x}^i} \log \nu(\boldsymbol{x}^i) + \sqrt{2\eta}\boldsymbol{\epsilon}^i \,, \tag{9}$$

with $\boldsymbol{\epsilon}^i \sim \mathcal{N}(\boldsymbol{0}, \boldsymbol{I})$. When $\eta \to 0$, the ULA converges to Langevin SDE, providing exact sampling.

Previous works (Durmus & Moulines, 2015; Cheng & Bartlett, 2018) show that ULA can efficiently converge to the target measure $\nu$ if $\nu$ is log-concave and smooth. However, when facing complex and multimodal distributions, we can only guarantee convergence to the concave vicinity.

## C.2 ANNEALED LANGEVIN MCMC SAMPLING

Langevin MCMC have been used to perform implicit sampling in energy-based models (Du & Mordatch, 2019) and score-based models (Song & Ermon, 2020). However, these methods suffer from inaccurate score estimation and low density regions (Song & Ermon, 2020). In Song & Ermon (2020) they propose to perturb the data with gaussian noise, eventually smoothing the data distribution:

$$q(\boldsymbol{x}_t) = \int_{\boldsymbol{x}_0} p_0(\boldsymbol{x}_0)\mathcal{N}(\boldsymbol{x}_t; \boldsymbol{x}_0, \sigma_t^2 \boldsymbol{I}) \,,$$

and creating a sequence of annealed distributions $\{q(\boldsymbol{x}_t)\}_{t=0}^T$ which converges to $p_0(\boldsymbol{x}_0)$. Since they are smoothed by gaussian noise, we can improve the mixing time of Langevin MCMC on multimodal distributions by sampling from these intermediate distributions, sharing similar spirits with simulated annealing (Kirkpatrick et al., 1983).

In Du et al. (2023), they extend this method to compositional generation of diffusion models. Specifically, we consider sampling from a product distribution $p_0^{\mathrm{prod}}(\boldsymbol{x}_0) \propto p_0^1(\boldsymbol{x}_0)p_0^2(\boldsymbol{x}_0)$, where $p_0^1(\boldsymbol{x}_0)$ and $p_0^2(\boldsymbol{x}_0)$ are distributions of different diffusion models. Since we have access to the score functions $\nabla_{\boldsymbol{x}_t} \log q_t^1(\boldsymbol{x}_t)$ and $\nabla_{\boldsymbol{x}_t} \log q_t^2(\boldsymbol{x}_t)$ through the diffusion model, we can construct a sequence of annealing distributions $\tilde{q}_t^{\mathrm{prod}}(\boldsymbol{x}_t)$ such that:

$$\nabla_{\boldsymbol{x}_t} \log \tilde{q}_t^{\mathrm{prod}}(\boldsymbol{x}_t) = \nabla_{\boldsymbol{x}_t} \log q_t^1(\boldsymbol{x}_t) + \nabla_{\boldsymbol{x}_t} \log q_t^2(\boldsymbol{x}_t) \,.$$

By sampling from the sequence $\left\{\tilde{q}_t^{\mathrm{prod}}(\boldsymbol{x}_t)\right\}$, we can arrive at $\tilde{q}_0^{\mathrm{prod}}(\boldsymbol{x}_0)$ which is equal to $p_0^{\mathrm{prod}}(\boldsymbol{x}_0)$. A key difference from sampling from $\left\{\tilde{q}_t^{\mathrm{prod}}(\boldsymbol{x}_t)\right\}$ and direct diffusion sampling is that the diffusion process with $p_0^{\mathrm{prod}}(\boldsymbol{x}_0)$ defined as

$$q_t^{\mathrm{prod}}(\boldsymbol{x}_t) = \int_{\boldsymbol{x}_0} p_0^{\mathrm{prod}}(\boldsymbol{x}_0)q(\boldsymbol{x}_t|\boldsymbol{x}_0)$$

is different from $\tilde{q}_t^{\mathrm{prod}}(\boldsymbol{x}_t)$. The score of $q_t^{\mathrm{prod}}(\boldsymbol{x}_t)$ can be derived as:

$$\nabla_{\boldsymbol{x}_t} \log q_t^{\mathrm{prod}}(\boldsymbol{x}_t) = \nabla_{\boldsymbol{x}_t} \log \left( \int_{\boldsymbol{x}_0} p_0^1(\boldsymbol{x}_0)p_0^2(\boldsymbol{x}_0)q(\boldsymbol{x}_t|\boldsymbol{x}_0) \right) \,,$$

which is not equal to

$$\nabla_{\boldsymbol{x}_t} \log \tilde{q}_t^{\text{prod}}(\boldsymbol{x}_t) = \nabla_{\boldsymbol{x}_t} \log \left( \int_{\boldsymbol{x}_0} p_0^1(\boldsymbol{x}_0) q(\boldsymbol{x}_t|\boldsymbol{x}_0) \right) + \nabla_{\boldsymbol{x}_t} \log \left( \int_{\boldsymbol{x}_0} p_0^2(\boldsymbol{x}_0) q(\boldsymbol{x}_t|\boldsymbol{x}_0) \right) ,$$

and thus intractable to compute directly.

A key distinction between annealed Langevin MCMC sampling and reverse diffusion sampling is that we run multiple Langevin MCMC steps on the *same noise level*, while reverse diffusion goes from high noise level to low noise level via denoising. A minimal pseudo code is shown in Alg. 1.

---

**Algorithm 1** Annealed Langevin MCMC sampling

---

**Input**: sequence of annealing distributions $\{\tilde{q}_t(\boldsymbol{x}_t)\}_{t=0}^T$, number of MCMC steps $N$, step size $\{\eta_t\}_{t=0}^T$. (Optional) reverse transition kernel $\{\tilde{p}_\theta(\boldsymbol{x}_{t-1}|\boldsymbol{x}_t)\}_{t=0}^T$.
**Init**: $\boldsymbol{x}_T^0 \sim \mathcal{N}(\boldsymbol{0}, \boldsymbol{I})$
**for** $t = T, \cdots, 1$ **do**
  **for** $i = 0, 1, \cdots, N-1$ **do**
    Perform Langevin MCMC steps:

$$\boldsymbol{x}_t^{i+1} = \boldsymbol{x}_t^i + \eta_t \nabla_{\boldsymbol{x}_t} \log \tilde{q}_t(\boldsymbol{x}_t^i) + \sqrt{2\eta_t} \boldsymbol{\epsilon}_t^i, \quad \boldsymbol{\epsilon}_t^i \sim \mathcal{N}(\boldsymbol{0}, \boldsymbol{I}) .$$

  **end for**
  (Optional) transit to next time step: $\boldsymbol{x}_{t-1}^0 \sim \tilde{p}_\theta(\cdot|\boldsymbol{x}_t^N)$. If no reverse kernel initialize $\boldsymbol{x}_{t-1}^0 = \boldsymbol{x}_t^N$.
**end for**
Return $\boldsymbol{x}_0$

---

### C.3 ANNEALED LANGEVIN MCMC WITH RECURRENT TRAINING-FREE GUIDANCE

In this section, we prove the connection between annealed Langevin MCMC (Alg. 1) and training-free guidance (Alg. 2) in Proposition. 1. We divide the proof into two parts. In Sec. C.3.1 we prove the equivalence between naive recurrence steps and Langevin MCMC. Then in Sec. C.3.2, we prove that adding the guidance term is defining an annealing path that biases towards high score regions. Finally, we provide a rigorous convergence analysis in Sec. C.3.3.

### C.3.1 EQUIVALENCE BETWEEN LANGEVIN MCMC AND NAIVE RECURRENCE

Consider the diffusion process with the following stochastic interpolant (Ma et al., 2024):

$$\boldsymbol{x}_t = \alpha_t \boldsymbol{x}_0 + \sigma_t \boldsymbol{\epsilon} .$$

We denote the score function of $q_t(\boldsymbol{x}_t)$ as $\nabla_{\boldsymbol{x}_t} \log q_t(\boldsymbol{x}_t) = s(\boldsymbol{x}_t, t)$. Recall the forward process in Eq. 1:

$$\boldsymbol{x}_t = \frac{\alpha_t}{\alpha_{t-1}} \boldsymbol{x}_{t-1} + \sqrt{\alpha_t^2 \left( \frac{\sigma_t^2}{\alpha_t^2} - \frac{\sigma_{t-1}^2}{\alpha_{t-1}^2} \right)} \boldsymbol{\epsilon} . \tag{10}$$

In a recurrence step in Line. 5, we first solve $\boldsymbol{x}_{t-1}^i$ from $\boldsymbol{x}_t^i$ using the learned score function $s(\boldsymbol{x}_t^i, t)$, then add noise to $\boldsymbol{x}_{t-1}^i$ to obtain the recurrent sample $\boldsymbol{x}_t^{i+1}$, where the superscript denotes the recurrence step index: $i = 0, 1, \cdots, N_{\text{recur}}$. Depending on different solvers, we have different formulations of $\boldsymbol{x}_t^{i+1}$.

**DDIM sampler**. When using DDIM (Song et al., 2020a) sampler, we have the reverse step as:

$$\boldsymbol{x}_{t-1} = \frac{\alpha_{t-1}}{\alpha_t} \boldsymbol{x}_t + \sigma_t^2 \left( \frac{\alpha_{t-1}}{\alpha_t} - \frac{\sigma_{t-1}}{\sigma_t} \right) s(\boldsymbol{x}_t, t) , \tag{11}$$

where $s(\boldsymbol{x}_t, t)$ is the score function $\nabla_{\boldsymbol{x}_t} \log q_t(\boldsymbol{x}_t)$. Thus, we have:

$$\boldsymbol{x}_t^{i+1} = \frac{\alpha_t}{\alpha_{t-1}} \boldsymbol{x}_{t-1}^i + \alpha_t \sqrt{\frac{\sigma_t^2}{\alpha_t^2} - \frac{\sigma_{t-1}^2}{\alpha_{t-1}^2}} \boldsymbol{\epsilon}^i$$

$$= \boldsymbol{x}_t^i + \sigma_t^2 \left( 1 - \frac{\alpha_t}{\alpha_{t-1}} \frac{\sigma_{t-1}}{\sigma_t} \right) s(\boldsymbol{x}_t^i, t) + \sigma_t \sqrt{1 - \frac{\alpha_t^2}{\alpha_{t-1}^2} \frac{\sigma_{t-1}^2}{\sigma_t^2}} \boldsymbol{\epsilon}^i .$$

Denote $\lambda_t = \log \frac{\alpha_t}{\sigma_t}$, then we have:

$$
\begin{aligned}
\boldsymbol{x}_t^{i+1} &= \boldsymbol{x}_t^i + \sigma_t^2 \left(1 - e^{\lambda_t - \lambda_{t-1}}\right) s(\boldsymbol{x}_t^i, t) + \sigma_t \sqrt{1 - e^{2(\lambda_t - \lambda_{t-1})}} \boldsymbol{\epsilon}^i \\
&= \boldsymbol{x}_t^i + \sigma_t^2 \left(1 - e^{\lambda_t - \lambda_{t-1}}\right) s(\boldsymbol{x}_t^i, t) + \sigma_t \sqrt{\left(1 - e^{\lambda_t - \lambda_{t-1}}\right)\left(1 + e^{\lambda_t - \lambda_{t-1}}\right)} \boldsymbol{\epsilon}^i ,
\end{aligned}
$$

where $1 + e^{\lambda_t - \lambda_{t-1}} \to 2$ when $T \to \infty$ and denoising step size approaches 0, as $\lambda_t - \lambda_{t-1} \to 0$.

**DDPM sampler**. In DDPM (Ho et al., 2020), we parametrize the posterior distribution as:

$$
p_\theta(\boldsymbol{x}_{t-1}|\boldsymbol{x}_t) = \mathcal{N}\left(\boldsymbol{x}_{t-1}; \mu_\theta(\boldsymbol{x}_t, t), \Sigma_\theta(\boldsymbol{x}_t, t)\right) , \tag{12}
$$

where the posterior mean is:

$$
\mu_\theta(\boldsymbol{x}_t, t) = \frac{\alpha_{t-1}}{\alpha_t} \boldsymbol{x}_t + \left(\sigma_t^2 \frac{\alpha_{t-1}}{\alpha_t} - \sigma_{t-1}^2 \frac{\alpha_t}{\alpha_{t-1}}\right) s(\boldsymbol{x}_t, t) .
$$

Ho et al. (2020) parameterizes the posterior variance as $\Sigma_\theta(\boldsymbol{x}_t, t) = \beta_t \boldsymbol{I}$ or $\Sigma_\theta(\boldsymbol{x}_t, t) = \tilde{\beta}_t \boldsymbol{I}$:

$$
\beta_t = \alpha_t^2 \left(\frac{\sigma_t^2}{\alpha_t^2} - \frac{\sigma_{t-1}^2}{\alpha_{t-1}^2}\right) ,
$$

$$
\tilde{\beta}_t = \frac{\sigma_{t-1}^2}{\sigma_t^2} \beta_t ,
$$

while Nichol & Dhariwal (2021) propose to train the posterior variance as $\Sigma_\theta(\boldsymbol{x}_t, t) = \exp\left(v \log \beta_t + (1 - v) \log \tilde{\beta}_t\right)$.

Thus, a backward step can be written as:

$$
\boldsymbol{x}_{t-1} = \frac{\alpha_{t-1}}{\alpha_t} \boldsymbol{x}_t + \left(\sigma_t^2 \frac{\alpha_{t-1}}{\alpha_t} - \sigma_{t-1}^2 \frac{\alpha_t}{\alpha_{t-1}}\right) s(\boldsymbol{x}_t, t) + \Sigma_\theta^{1/2}(\boldsymbol{x}_t, t) \boldsymbol{\epsilon}_{\text{post}} ,
$$

where $\boldsymbol{\epsilon}_{\text{post}}$ denotes the noise added in the posterior sampling step. Then, we can write the recurrence step as:

$$
\begin{aligned}
\boldsymbol{x}_t^{i+1} &= \boldsymbol{x}_t^i + \left(\sigma_t^2 - \sigma_{t-1}^2 \frac{\alpha_t^2}{\alpha_{t-1}^2}\right) s(\boldsymbol{x}_t^i, t) + \Sigma_\theta^{1/2}(\boldsymbol{x}_t^i, t) \boldsymbol{\epsilon}_{\text{post}}^i + \alpha_t \sqrt{\frac{\sigma_t^2}{\alpha_t^2} - \frac{\sigma_{t-1}^2}{\alpha_{t-1}^2}} \boldsymbol{\epsilon}_{\text{forward}}^i \\
&= \boldsymbol{x}_t^i + \beta_t s(\boldsymbol{x}_t^i, t) + \sqrt{\Sigma_\theta(\boldsymbol{x}_t, t) + \beta_t \boldsymbol{I}} \boldsymbol{\epsilon}^i ,
\end{aligned}
$$

where $\Sigma_\theta(\boldsymbol{x}_t, t) \to \beta_t \boldsymbol{I}$ when $T \to \infty$, and the denoising step size approaches 0.

**Flow-Matching sampler**. In flow-matching models (Ma et al., 2024; Esser et al., 2024), the noise schedule is $\alpha_t = 1 - t, \sigma_t = t$, with $t \in [0, 1]$. The denoising step can be written as:

$$
\begin{aligned}
\boldsymbol{x}_{t-\Delta t} &= \boldsymbol{x}_t - \boldsymbol{v}_\theta(\boldsymbol{x}_t, t) \Delta t \\
&= \boldsymbol{x}_t - \frac{-t s(\boldsymbol{x}_t, t) - \boldsymbol{x}_t}{1 - t} \Delta t \\
&= (1 + \frac{\Delta t}{1 - t}) \boldsymbol{x}_t + \frac{t \Delta t}{1 - t} s(\boldsymbol{x}_t, t)
\end{aligned}
$$

Then, the forward process in Eq. 10 can be written as:

$$
x_t = \frac{1 - t}{1 - (t - \Delta t)} \boldsymbol{x}_{t-\Delta t} + \frac{\sqrt{\Delta t \left(2t - \Delta t - 2t^2 + 2t\Delta t\right)}}{1 - t + \Delta t} \boldsymbol{\epsilon} .
$$

Thus, the recurrence equation can be written as:

$$
\begin{aligned}
\boldsymbol{x}_t^{i+1} &= \boldsymbol{x}_t^i + \frac{t \Delta t}{1 - t + \Delta t} s(\boldsymbol{x}_t, t) + \frac{\sqrt{\Delta t \left(2t - \Delta t - 2t^2 + 2t\Delta t\right)}}{1 - t + \Delta t} \boldsymbol{\epsilon}^i \\
&= \boldsymbol{x}_t^i + \frac{t \Delta t}{1 - t + \Delta t} s(\boldsymbol{x}_t, t) + \frac{\sqrt{2t\Delta t \left(1 - t + \Delta t\right) - \Delta t^2}}{1 - t + \Delta t} \boldsymbol{\epsilon}^i
\end{aligned}
$$

**Putting together**. In general, we can write the recurrence step as:

$$
\boldsymbol{x}_t^{i+1} = \boldsymbol{x}_t^i + a_t r_t s(\boldsymbol{x}_t^i, t) + \sqrt{2a_t} \boldsymbol{\epsilon}^i . \tag{13}
$$

with $a_t \to 0$ and $r_t \to 1$ as the denoising step size approaches 0:

- For DDIM sampler, we have $a_t = \frac{1}{2}\alpha_t^2\left(\frac{\sigma_t^2}{\alpha_t^2} - \frac{\sigma_{t-1}^2}{\alpha_{t-1}^2}\right)$ and $r_t = \frac{2}{1+e^{\lambda_t - \lambda_{t-1}}}$.

- For DDPM sampler, we have $a_t = \frac{1}{2}\alpha_t^2\left(\frac{\sigma_t^2}{\alpha_t^2} - \frac{\sigma_{t-1}^2}{\alpha_{t-1}^2}\right)$ and $1 \le r_t \le \frac{2}{1 + \frac{\sigma_{t-1}^2}{\sigma_t^2}}$.

- For flow-matching sampler, we have $a_t = \frac{t\Delta t}{1-t}$ and $1 \le r_t \le 1 + \frac{\Delta t}{t(1-t)}$

Thus, it can be seen as a approximation of the ULA in Eq. 9, and also a discretization of the Langevin SDE in Eq. 8.

### C.3.2 ANNEALED LANGEVIN MCMC WITH GUIDANCE

When applying training free guidance (Ye et al., 2024) during the recurrence, we have:
$$\boldsymbol{x}_t^{i+1} = \boldsymbol{x}_t^i + a_t r_t s(\boldsymbol{x}_t^i, t) + \sqrt{2a_t}\boldsymbol{\epsilon}^i + \boldsymbol{\Delta}(\boldsymbol{x}_t, t),$$
where $a_t$, $b_t$ are the coefficients of the recurrence equation in Eq. 13 without guidance. In general, $\boldsymbol{\Delta}_t = \rho_t \nabla_{\boldsymbol{x}_t} \log f(\boldsymbol{x}_{0|t}) + \mu_t \alpha_t \nabla_{\boldsymbol{x}_{0|t}} \log f(\boldsymbol{x}_{0|t})$, where $\rho_t, \mu_t$ controls the guidance strength. We then show that the guidance term can be considered as the score function of a set of annealed verifiers $\left\{\hat{f}(\boldsymbol{x}_t)\right\}_{t=0}^T$.

When considering 'variance guidance' in Line. 7, we have $\boldsymbol{\Delta}_{\mathrm{var}} = \rho_t \nabla_{\boldsymbol{x}_t} \log f(\boldsymbol{x}_{0|t})$. Thus, we can define $\hat{f}_t^{\mathrm{var}}(\boldsymbol{x}_t) = f(\boldsymbol{x}_{0|t})$, which satisfies $\hat{f}_0^{\mathrm{var}}(\boldsymbol{x}_0) = f(\boldsymbol{x}_0)$. Similarly, for 'mean guidance' in Line. 8, we have

$$\boldsymbol{\Delta}_{\mathrm{mean}} = \mu_t \alpha_t \nabla_{\boldsymbol{x}_{0|t}} \log f(\boldsymbol{x}_{0|t})$$

$$= \mu_t \frac{\sigma_t^2}{\boldsymbol{\Sigma}_{0|t}} \nabla_{\boldsymbol{x}_t} \log f(\boldsymbol{x}_{0|t}),$$

where the second Equation follows from Lemma 3.3 in Ye et al. (2024). Thus, there exists a set of functions $\hat{f}_t^{\mathrm{mean}}(\boldsymbol{x}_t)$ such that $\nabla_{\boldsymbol{x}_t} \log \hat{f}_t^{\mathrm{mean}}(\boldsymbol{x}_t) = \frac{\sigma_t^2}{\boldsymbol{\Sigma}_{0|t}} \nabla_{\boldsymbol{x}_t} \log f(\boldsymbol{x}_{0|t})$, and we can see that when $t \to 0, \nabla_{\boldsymbol{x}_t} \log \hat{f}_t^{\mathrm{mean}}(\boldsymbol{x}_t) = \nabla_{\boldsymbol{x}_0} \log f(\boldsymbol{x}_0)$. If we additionally incorporate the 'implicit dynamics' in Line. 4, our arguments still stands since the smoothed objective $\tilde{f}(\boldsymbol{x}) = \mathbb{E}_{\boldsymbol{\delta} \sim \mathcal{N}(\boldsymbol{0}, \boldsymbol{I})} f(\boldsymbol{x} + \bar{\gamma}\sigma_t \boldsymbol{\delta})$ converges to $f$ with $t \to 0$ and $\sigma_t \to 0$.

Combining the two terms together, we have $\boldsymbol{\Delta}_t = c_t \nabla_{\boldsymbol{x}_t} \log \hat{f}_t(\boldsymbol{x}_t)$ with $\hat{f}_t = \hat{f}_t^{\mathrm{var}} \cdot \hat{f}_t^{\mathrm{mean}}$. Thus, recurrence with guidance can be written as:

$$\boldsymbol{x}_t^{i+1} = \boldsymbol{x}_t^i + a_t r_t s(\boldsymbol{x}_t^i, t) + \sqrt{2a_t}\boldsymbol{\epsilon}^i + c_t \nabla_{\boldsymbol{x}_t} \log \hat{f}_t(\boldsymbol{x}_t)$$

$$= \boldsymbol{x}_t^i + a_t r_t \nabla_{\boldsymbol{x}_t} \log q_t(\boldsymbol{x}_t) \hat{f}_t(\boldsymbol{x}_t)^{c_t/a_t r_t} + \sqrt{2a_t}\boldsymbol{\epsilon}^i,$$

Thus, we have defined the annealing path as $\tilde{q}_t(\boldsymbol{x}_t) = q_t(\boldsymbol{x}_t)\hat{f}_t(\boldsymbol{x}_t)^{c_t/a_t r_t}$, $t = 1, 2, \cdots, T$.

### C.3.3 CONVERGENCE ANALYSIS

In this section, we provide a rigorous convergence analysis of recurrence to the target distribution $\tilde{q}_t(\boldsymbol{x}_t)$.

**Theorem 1.** *Suppose $\tilde{q}_t(\boldsymbol{x}_t)$ has bounded support, is $\alpha$-strongly log-concave and $L$-log-smooth, and $-\nabla^2 \log \tilde{q}_t$ is $M$-Lipschitz. Denote $\boldsymbol{x}_t^{N_{recur}}$ as the sample after $N_{recur}$ steps of recurrence, we can bound the Wasserstein distance between the distribution of $\boldsymbol{x}_t^{N_{recur}}$ and $\tilde{q}_t$ as:*

$$W_2(p(\boldsymbol{x}_t^{N_{recur}}), \tilde{q}_t) = \mathcal{O}\left(\sqrt{\lambda_{t-1} - \lambda_t} + e^{-2\lambda_t} - e^{-2\lambda_{t-1}} + (1 - e^{-2\lambda_t} + e^{-2\lambda_{t-1}})^{N_{recur}}\right),$$

*where $\lambda_t = \log\frac{\alpha_t}{\sigma_t}$ is half of the log SNR.*

*Proof.* Recall recurrence is equivalent to the following recursion equation, with $a_t$, $r_t$ being sampler-dependent parameters in Eq. 13:

$$\boldsymbol{x}_t^{i+1} = \boldsymbol{x}_t^i + a_t r_t \nabla_{\boldsymbol{x}_t} \log \tilde{q}_t(\boldsymbol{x}_t^i) + \sqrt{2a_t}\boldsymbol{\epsilon}^i$$

$$= \boldsymbol{x}_t^i + a_t \nabla_{\boldsymbol{x}_t} \log \tilde{q}_t(\boldsymbol{x}_t^i)^{r_t} + \sqrt{2a_t}\boldsymbol{\epsilon}^i.$$

Thus, recurrence is equivalent to running unadjusted Langevin algorithm (ULA) on the tempered distribution $p^{\text{tempered}} \propto \tilde{q}_t^{r_t}$. Using Lemma 1 and Lemma 2 from Wibisono (2018), given the regularity conditions on $\tilde{q}_t$, we can bound the discretization error from ULA as:

$$
\begin{aligned}
W_2(p^{\text{tempered}}, p(\boldsymbol{x}_t^{N_{\text{recur}}})) &= \mathcal{O}\left(a_t + (1 - a_t)^{N_{\text{recur}}}\right) \\
&= \mathcal{O}\left(\frac{\sigma_t^2}{\alpha_t^2} - \frac{\sigma_{t-1}^2}{\alpha_{t-1}^2} + (1 - \frac{\sigma_t^2}{\alpha_t^2} + \frac{\sigma_{t-1}^2}{\alpha_{t-1}^2})^{N_{\text{recur}}}\right) \\
&= \mathcal{O}\left(e^{-2\lambda_t} - e^{-2\lambda_{t-1}} + (1 - e^{-2\lambda_t} + e^{-2\lambda_{t-1}})^{N_{\text{recur}}}\right) .
\end{aligned}
$$

To bound $W_2(p^{\text{tempered}}, \tilde{q}_t)$, we first bound the TV distance as $\text{TV}(p^{\text{tempered}}, \tilde{q}_t)$.

Denote $Z(r_t) = \int \tilde{q}_t(x)^{r_t} dx$, and $\psi(r) = \log Z(r)$. We have:

$$
\begin{aligned}
&\text{KL}(p^{\text{tempered}}, \tilde{q}_t) \\
&= \mathbb{E}_{p^{\text{tempered}}}\left[r_t \log \tilde{q}_t - \psi(r_t) - \log \tilde{q}_t\right] \\
&\leq \mathcal{O}\left((r_t - 1)\mathbb{E}_{p^{\text{tempered}}}\left[\log \tilde{q}_t - \psi'(r)\right] + \psi''(r)(r_t - 1)^2\right) ,
\end{aligned}
$$

where the last step is using Taylor expansion. Since $\psi'(r) = \mathbb{E}_{p^{\text{tempered}}}[\log \tilde{q}_t]$ and $\psi''(r) = \text{Var}_{p^{\text{tempered}}}[\log \tilde{q}_t]$. Using the Pinsker inequality, we have $\text{TV}(p^{\text{tempered}}, \tilde{q}_t) \leq \frac{1}{2}\sqrt{\text{KL}(p^{\text{tempered}}, \tilde{q}_t)} \leq \mathcal{O}\left((r_t - 1)\sqrt{\text{Var}_{p^{\text{tempered}}}[\log \tilde{q}_t]}\right) = \mathcal{O}(r_t - 1)$, given the finiteness of $\text{Var}_{p^{\text{tempered}}}[\log \tilde{q}_t]$ under bounded support.

Following Proposition 7.10 in Villani (2003) for distributions with bounded support, we have:

$$
\begin{aligned}
&W_2(p^{\text{tempered}}, \tilde{q}_t) \\
&= \mathcal{O}\left(\sqrt{\text{TV}(p^{\text{tempered}}, \tilde{q}_t)}\right) \\
&= \mathcal{O}\left(\sqrt{r_t - 1}\right) \\
&= \mathcal{O}\left(\sqrt{1 - \min\left(\frac{\alpha_t \sigma_{t-1}}{\alpha_{t-1}\sigma_t}, \frac{\sigma_{t-1}^2}{\sigma_t^2}\right)}\right) \\
&= \mathcal{O}\left(\sqrt{\log \frac{\sigma_t}{\sigma_{t-1}} + \max\left(\log \frac{\alpha_{t-1}}{\alpha_t}, \log \frac{\sigma_t}{\sigma_{t-1}}\right)}\right) \\
&= \mathcal{O}\left(\sqrt{\log \frac{\alpha_{t-1}}{\alpha_t} + \log \frac{\sigma_t}{\sigma_{t-1}}}\right) \\
&= \mathcal{O}\left(\sqrt{\lambda_{t-1} - \lambda_t}\right) ,
\end{aligned}
$$

where we recall the definition of $r_t$ in Eq. 13.

Putting together the bound on $W_2(p^{\text{tempered}}, p(\boldsymbol{x}_t^{N_{\text{recur}}}))$ and $W_2(p^{\text{tempered}}, \tilde{q}_t)$, we obtain our desired bound. $\qquad \square$

## C.4 RELATIONSHIP BETWEEN LANGEVIN MCMC AND GRADIENT ASCENT

In training-free guidance, most prior works only apply gradient ascent without recurrence. Here we provide a theoretical analysis of both methods.

Recall the KL-divergence objective in Eq. 6, which can be further decomposed when we are sampling from a compositional distribution of $p_0(\boldsymbol{x}_0)$ and verifier $f(\boldsymbol{x}_0)$, with $\nu \propto p_0 \cdot f$:

$$
H_\nu(\rho) = \mathbb{E}_\rho[-\log f] + H_{p_0}(\rho) + \log Z .
$$

where $Z = \int p_0 f$ is a normalization constant. Thus, gradient ascent is optimizing the verifier objective $\mathbb{E}_\rho[-\log f]$, while Langevin MCMC in Eq. 13 is optimizing the divergence between current sample and base distribution $H_{p_0}(\rho)$. This explains why naive gradient updates leads to OOD samples, and recurrence effectively mitigates this issue, acting as a contraction force pulling the

sample back to the original manifold. However, since we start from $p_0$ as the distribution of our initial sample, sometimes we can omit the recurrence if the guidance strength is small. But if we wish to traverse different modes with multiple gradient updates, introducing recurrence helps to avoid OOD during optimization.

## C.5 Implementing Local Search with TFG hyper-parameter space

Due to the equivalence between annealed Langevin MCMC and training-free guidance with recurrence, we can implement local search with Langevin MCMC using the TFG framework of Ye et al. (2024), efficiently searching the hyperparameters. Here we provide a overview of the algorithm and design space. Following Sec. C.3, every iteration of recurrence in Line. 5 is equivalent to an annealed Langevin MCMC step, thus $N_{\text{recur}}$ is equal to the number of local search steps.

---

**Algorithm 2** Training-Free Guidance

1: **Input:** Unconditional diffusion model $\epsilon_\theta$, verifier $f$, guidance strength $\rho, \mu, \bar\gamma$, number of steps $T, N_{\text{recur}}, N_{\text{iter}}$
2: $\boldsymbol{x}_T \sim \mathcal{N}(\mathbf{0}, \boldsymbol{I})$
3: **for** $t = T, \cdots, 1$ **do**
4:      Define function $\tilde{f}(\boldsymbol{x}) = \mathbb{E}_{\boldsymbol{\delta} \sim \mathcal{N}(\mathbf{0}, \boldsymbol{I})} f(\boldsymbol{x} + \bar\gamma \sigma_t \boldsymbol{\delta})$
5:      **for** $r = 1, \cdots, N_{\text{recur}}$ **do**
6:          $\boldsymbol{x}_{0|t} = (\boldsymbol{x}_t - \sigma_t \epsilon_\theta(\boldsymbol{x}_t, t))/\alpha_t$
7:          $\boldsymbol{\Delta}_{\text{var}} = \rho_t \nabla_{\boldsymbol{x}_t} \log \tilde{f}(\boldsymbol{x}_{0|t})$
8:          $\boldsymbol{\Delta}_{\text{mean}} = \boldsymbol{\Delta}_{\text{mean}} + \mu_t \alpha_t \nabla_{\boldsymbol{x}_{0|t}} \log \tilde{f}(\boldsymbol{x}_{0|t} + \boldsymbol{\Delta}_{\text{mean}})$     ▷*Iterate $N_{\text{iter}}$ times starting from $\boldsymbol{\Delta}_{mean} = \mathbf{0}$*
9:          $\boldsymbol{x}_{t-1} = \text{Sample}(\boldsymbol{x}_t, \boldsymbol{x}_{0|t}, t) + \frac{\alpha_{t-1}}{\alpha_t}(\boldsymbol{\Delta}_{\text{var}} + \boldsymbol{\Delta}_{\text{mean}})$     ▷ *Sample follows DDIM or DDPM*
10:          $\boldsymbol{x}_t \sim \mathcal{N}\left(\frac{\alpha_t}{\alpha_{t-1}} \boldsymbol{x}_{t-1}, \alpha_t^2 \left(\frac{\sigma_t^2}{\alpha_t^2} - \frac{\sigma_{t-1}^2}{\alpha_{t-1}^2}\right) \boldsymbol{I}\right)$     ▷ *Recurrent strategy*
11:      **end for**
12: **end for**
13: **Output:** Conditional sample $\boldsymbol{x}_0$

---

For time varying schedules $\rho_t, \mu_t$, we follow Ye et al. (2024) and propose to use either the 'increase' schedule:

$$s_t = T \frac{\alpha_t/\alpha_{t-1}}{\sum_{t=1}^T \alpha_t/\alpha_{t-1}}, \tag{14}$$

where we increase the guidance strength as we denoise: $s_T < s_{T-1} < \cdots < s_1$; or the 'constant' schedule

$$s_t = 1, \tag{15}$$

which uses constant parameters throughout the denoising process. Thus, the time-varying schedules can be computed as $\rho_t = s_t \bar\rho$ and $\mu_t = s_t \bar\mu$, and we only need to determine the average $\bar\rho$ and $\bar\mu$.

# D Global Search of Denoising Diffusion Models

In this section, we provide details about the global search algorithms: BFS and DFS.

## D.1 BFS-Based Search

We present the pseudo code for BFS in Alg. 3. In practice, we can evaluate and resample the particles at a fixed subset of time steps, with detailed hyper-parameters in the details of each experiment.

### D.1.1 Overview of design space

**Tempering**. When estimating the particle score via the denoised output $f(\boldsymbol{x}_{0|t}^k)$, for larger $t$ the estimation often has larger bias, due to the fact that $\mathbb{E}[f(\boldsymbol{x}_0)|\boldsymbol{x}_t] \neq f(\mathbb{E}[\boldsymbol{x}_0|\boldsymbol{x}_t]) = f(\boldsymbol{x}_{0|t})$. Thus, we propose to re-weight the score estimates with a tempering schedule $\tau_t$ before scoring the particles. For a increasing schedule $\tau_T < \tau_{T-1} < \cdots < \tau_1 < \tau_0$, the estimated scores have more influence at

---

**Algorithm 3** Diffusion BFS

---

**Diffusion input**: diffusion model $\epsilon_\theta$ with diffusion time steps $T$ and proposal transition kernel $\{\tilde{p}_\theta(\boldsymbol{x}_{t-1}|\boldsymbol{x}_t)\}_{t=1}^N$. Verifier $f$.
**BFS input**: Tempering schedule $\tau_t$. Budget of particles $N$. Scoring function and resampling method.
**Init**: Random sample $N$ particles $\boldsymbol{x}_T^k \sim \mathcal{N}(\boldsymbol{0}, \boldsymbol{I}), k = 1, 2, \cdots N$.
**for** $t = T, \cdots, 1$ **do**
    **for** $k = 1, 2, \cdots, N$ **do**
        **Estimation**. Estimate the conditional mean: $\boldsymbol{x}_{0|t}^k = \frac{\boldsymbol{x}_t^k - \sigma_t \epsilon_\theta(\boldsymbol{x}_t^k, t)}{\alpha_t}$. Compute the verifier score estimate $f(\boldsymbol{x}_{0|t}^k)$.
        **Scoring**. Score the particles according to the scoring functions:

$$\textbf{Current} : \widehat{f}(\boldsymbol{x}_t^k) = \tau_t f(\boldsymbol{x}_{0|t}^k),$$

$$\textbf{Difference} : \widehat{f}(\boldsymbol{x}_t^k) = \tau_t f(\boldsymbol{x}_{0|t}^k) - \widehat{f}_{\text{prev}}^k,$$

$$\textbf{Max} : \widehat{f}(\boldsymbol{x}_t^k) = \max\left(\tau_t f(\boldsymbol{x}_{0|t}^k), \widehat{f}_{\text{prev}}^k\right).$$

        **Resampling**. Compute the weights

$$(w_t^1, w_t^2, \cdots, w_t^N) = \text{softmax}\left(\widehat{f}(\boldsymbol{x}_t^1), \widehat{f}(\boldsymbol{x}_t^2), \cdots \widehat{f}(\boldsymbol{x}_t^N)\right),$$

        and resample the children

$$(n_t^1, n_t^2 \cdots, n_t^N) = \text{Resample}(N; w_t^1, w_t^2, \cdots, w_t^N),$$

        where Resample can be Multinomial resampling or SSP resampling (Alg.4).
    **end for**
    **for** $k = 1, \cdots, N$ **do**
        Sample $n_t^k$ children particles from $\boldsymbol{x}_t^k$: $\boldsymbol{x}_{t-1}^j \sim \tilde{p}_\theta(\cdot|\boldsymbol{x}_t^k), j = 1, 2, \cdots, n_t^k$
    **end for**
    Update the score buffers for computing the **Difference** and **Max** score at next timestep.

$$\widehat{f}_{\text{prev}}^k = \begin{cases} \widehat{f}\left(\boldsymbol{x}_t^{\text{parent(k)}}\right) & \text{if scoring function is } \textbf{Max} \\ \tau_t f\left(\boldsymbol{x}_{0|t}^{\text{parent(k)}}\right) & \text{if scoring function is } \textbf{Difference} \end{cases}$$

**end for**
**Return** $\boldsymbol{x}_0 = \text{argmax}_{\boldsymbol{x}_0^1, \cdots, \boldsymbol{x}_0^N} f(\boldsymbol{x}_0^k)$

---

smaller time steps with higher weights, where the estimation has less bias. For the increasing tempering schedule, we simply adopt the design in DAS (Kim et al., 2025) with $\tau_t = \left((1+\gamma)^{T-t} - 1\right)\tau$, with $\gamma$ searched in $\{0.008, 0.024\}$. We also consider the deterministic particle selection in SVDD (Li et al., 2024) where we only retain the highest scoring particle, which can be seen as $\tau_t = \infty$. For the constant tempering schedule we simply set $\tau_t = \tau$, without any tempering on different time steps.

**Scoring**. Given the score estimates $f(\boldsymbol{x}_{0|t})$, one can construct scoring rules based on the scores of the entire path $\left\{f(\boldsymbol{x}_{0|s}^k)\right\}_{s=T}^t$. In traditional sequential-monte-carlo (SMC) based methods, the score is computed as the difference between two consecutive evaluations: $\widehat{f}(\boldsymbol{x}_t^k) = \tau_t f(\boldsymbol{x}_{0|t}^k) - \tau_{t+1} f(\boldsymbol{x}_{0|t+1}^k)$. However, in FK-steering (Singhal et al., 2025a), they pointed out that the difference potential is unfair for sample path that saturates earlier, reaching a high reward at earlier time steps. Thus they propose the max scoring: $\widehat{f}(\boldsymbol{x}_t^k) = \max_{s \geq t} \tau_t f(\boldsymbol{x}_{0|s}^k)$. We also consider the simple baseline of current score: $\widehat{f}(\boldsymbol{x}_t^k) = \tau_t f(\boldsymbol{x}_{0|t}^k)$.

**Resampling**. Given the scores $\widehat{f}(\boldsymbol{x}_t^k)$, we can compute the softmax logits of particles via $\left\{w_t^k\right\}_{k=1}^N = \text{softmax}\left\{\widehat{f}(\boldsymbol{x}_t^k)\right\}_{k=1}^N$. Then, we resample the particles according to the logits, and obtain the number of children for each particle $n_t^k$. The sets $\left\{n_t^k\right\}_{k=1}^N$ should satisfy the following constraints:

$$\sum_{k=1}^N n_t^k = N$$
$$\mathbb{E}[n_t^k] = N w_t^k$$

Most previous approaches simply adopt the multinomial resampling, with $\left\{n_t^k\right\}_{k=1}^N \sim \text{Multinomial}\left(N; \left\{w_t^k\right\}_{k=1}^N\right)$. However, this sampling process introduces large variance since every component is sampled independently. To reduce variance, for example, residual resampling only resamples the residuals $\tilde{n}_t^k$ from $N - R$ with weights $\tilde{w}_t^k = \frac{Nw_t^k - [Nw_t^k]}{N-R}$, where $R = \sum_{k=1}^N [Nw_t^k]$. Then, we obtain $n_t^k = [Nw_t^k] + \tilde{n}_t^k$. This way we reduce the randomness in $n_t^k$.

The SSP resampling views the sampling process as randomized rounding on the expectations $Nw_t^k$, with pseudo code from Gerber et al. (2020) shown below in Alg. 4.

---

**Algorithm 4** SSP resampling

---

**Inputs:** $u \in [0,1]^N$ and $(\xi_1, \ldots, \xi_N) \in \mathbb{R}_+^N$ such that $\xi_n = Nw_t^n$ with $w_t^n$ being the softmax weight of the $n$-th particle, and we have $\sum_{n=1}^N \xi_n = N \in \mathbb{N}$.

**Output:** $\left(Y_{\text{ssp}}^1(u), \ldots, Y_{\text{ssp}}^N(u)\right) \in \mathbb{N}^N$ such that $\sum_{n=1}^N Y_{\text{ssp}}^n(u) = \sum_{n=1}^N \xi_n$.

Initialization: $\left(Y_{\text{ssp}}^1(u), \ldots, Y_{\text{ssp}}^N(u)\right) \leftarrow (\xi_1, \ldots, \xi_N), (n, m, k) \leftarrow (1, 2, 1)$

Iterate the following steps until $\left(Y_{\text{ssp}}^1(u), \ldots, Y_{\text{ssp}}^N(u)\right) \in \mathbb{N}^N$:

(1) Let $\delta$ be the smallest number in $(0,1)$ such that at least one of $Y_{\text{ssp}}^n(u) + \delta$ or $Y_{\text{ssp}}^m(u) - \delta$ is an integer, and let $\epsilon$ be the smallest number in $(0,1)$ such that at least one of $Y_{\text{ssp}}^n(u) - \epsilon$ or $Y_{\text{ssp}}^m(u) + \epsilon$ is an integer.

(2) If $u_k \leq \epsilon/(\delta + \epsilon)$ set $(Y_{\text{ssp}}^n(u), Y_{\text{ssp}}^m(u)) \leftarrow (Y_{\text{ssp}}^n(u) + \delta, Y_{\text{ssp}}^m(u) - \delta)$; otherwise set $(Y_{\text{ssp}}^n(u), Y_{\text{ssp}}^m(u)) \leftarrow (Y_{\text{ssp}}^n(u) - \epsilon, Y_{\text{ssp}}^m(u) + \epsilon)$.

(3) Update $n$ and $m$ as follows:

    1. If $\left(Y_{\text{ssp}}^n(u), Y_{\text{ssp}}^m(u)\right) \in \mathbb{N}^2, (n, m) \leftarrow (m+1, m+2)$;

    2. If $Y_{\text{ssp}}^n(u) \in \mathbb{N}$ and $Y_{\text{ssp}}^m(u) \notin \mathbb{N}$ set $(n, m) \leftarrow (m, m+1)$;

    3. if $Y_{\text{ssp}}^n(u) \notin \mathbb{N}$ and $Y_{\text{ssp}}^m(u) \in \mathbb{N}$ set $(n, m) \leftarrow (n, m+1)$.

(4) $k \leftarrow k + 1$

---

There are more variance reduced sampling methods such as systematic resampling and stratified resampling, and we adopt the Srinivasan sampling process (SSP) resampling as our design choice. For other variance-reduced methods, the performance is similar to SSP resampling, all outperforming naive multinomial resampling.

### D.1.2 OVERVIEW OF PRIOR METHODS

Here, we provide an overview of prior methods.

**SVDD (Li et al., 2024)**. In SVDD, the best sample is selected at each time step, from which $M$ children are generated. This approach can be viewed as a variant of BFS with $\tau = \infty$ and $M$ particles. Nodes are evaluated using the current score estimate $f(\boldsymbol{x}_{0|t})$.

**TreeG (Guo et al., 2025)**. In TreeG, particles are ranked and the top $M$ are either selected directly or resampled based on their scores to obtain $M$ samples. From each selected particle, $K$ children are

sampled, resulting in an effective tree width of $KM$. Particles are evaluated using their current score $f(\boldsymbol{x}_{0|t})$. They also consider adding Gaussian noise to $\boldsymbol{x}_{0|t}$ to approximate the posterior distribution $p(\boldsymbol{x}_0|\boldsymbol{x}_t)$. However, the posterior distribution is intractable as it is not a Gaussian distribution, and exact Gaussian approximation requires computing the variance using the Jacobian of the score function (see Ye et al. (2024)).

**DAS (Kim et al., 2025).** In DAS, the authors propose an exponentially increasing tempering schedule as the default, given by $\tau_t = (1 + \gamma)^{T-t} - 1$, and also introduce an adaptive tempering schedule. They adopt advanced SSP resampling instead of multinomial resampling, and evaluate particles based on the difference in rewards from the previous evaluation, similar to SMC methods. In the original implementation, DAS also modify the transition kernel with verifier gradients $\tau = \mathcal{N}\left(\boldsymbol{x}_{t-1}; \mu_\theta(\boldsymbol{x}_t) + \lambda \nabla_{\boldsymbol{x}_t} \log f(\boldsymbol{x}_{0|t})\right)$, and include the log ratio in the resampling weights: $w_t^i = \log p_\theta(\boldsymbol{x}_{t-1}^i|\boldsymbol{x}_t^i) - \log \tau(\boldsymbol{x}_{t-1}^i|\boldsymbol{x}_t^i) + \lambda \log f(\boldsymbol{x}_t^i) - \lambda \log f(\boldsymbol{x}_{t-1}^i)$. For a fair comparison, we disable the gradients in the BFS experiments in Table. 2, and provide additional results in Table. 7.

**FK-steering (Singhal et al., 2025a).** In FK, the authors propose several options for evaluating intermediate particles, including difference, max, and sum, with *max* adopted as the default. In the official implementation, multinomial resampling is used, which may lead to suboptimal performance.

**SoP (Ma et al., 2025).** In search-over-paths (SoP), we start with $N$ samples and denoise them to a low noise level $\sigma$. After that, we sample $M$ i.i.d noise for each of the noisy particles and run the forward process from $\sigma$ to $\sigma + \Delta f$, and then denoise from $\sigma + \Delta f$ to $\sigma + \Delta f - \Delta_b$. After that we select the top $N$ particles and repeat the above process until $\sigma = 0$. Although the noise injection is similar to the backtracking in DFS, we point out that SoP adopts a fixed schedule and all particles remain at the same noise level, which is different from our adaptive DFS.

**DSearch (Li et al., 2025b).** In DSearch, the authors improve upon beam-search based tree search by dynamically adjust the beam-size and tree width during sampling. At lower noise levels, we adopt larger beam-size and lower tree width, facilitating more local exploration. This approach improves the diversity of the generated samples.

### D.2 DFS-BASED SEARCH

In this section, we provide the details and pseudo code for DFS in Alg. 5. To better utilize previously explored sampling paths, we employ a buffer to store prior results. When the budget is exhausted and no more backtracking is allowed, we retrieve the best sample from the buffer.

Similar to BFS, controlling the set of evaluation steps allows a trade-off between efficiency and accuracy. Evaluating at earlier time steps introduces higher uncertainty but enables backtracking. Additionally, adjusting the backtracking depth $\Delta_T$ governs the search scope: a small $\Delta_T$ reduces computation and favors local search, while a larger $\Delta_T$ enables broader exploration at the cost of increased computation.

In practice, we set the evaluation steps to $\mathcal{S} = \left\{\frac{1}{2}T, \frac{1}{4}T\right\}$ for image experiments to save compute, and to $\mathcal{S} = \left\{\frac{3}{4}T, \frac{3}{4}T - 1, \cdots, 1\right\}$ for PointMaze experiments. We set the recurrence depth to $T/2$ for image tasks and $T/4$ for PointMaze, corresponding to the denoised steps at which samples are first evaluated. The threshold schedule $\delta_t$ is also set to 'increase' as in Eq.14 in our PointMaze experiments, enforcing tighter constraints for samples with lower noise. We use a simple constant $\delta_t = \delta$ schedule in our image experiments.

In our experiments, we observed that when backtracking to $t_{\text{next}} = T$—thus fully restarting—the nonzero terminal SNR $\alpha_T/\sigma_T$ in many diffusion schedules (Lin et al., 2024) can lead to cumulative errors with repeated backtracking. Therefore, when backtracking to $t_{\text{next}} = T$, we initialize with fresh Gaussian noise.

---

**Algorithm 5** Diffusion DFS

---

**Diffusion input**: diffusion model $\epsilon_\theta$ with diffusion time steps $T$ and proposal transition kernel $\{\tilde{p}_\theta(\boldsymbol{x}_{t-1}|\boldsymbol{x}_t)\}_{t=1}^N$. Verifier $f$.

**DFS input**: Budget for total number of backtracking $B = K$, backtracking depth $\Delta_T$ and threshold $\{\delta_t\}_{t=1}^T$.

**Init** $\boldsymbol{x}_T \sim \mathcal{N}(\boldsymbol{0}, \boldsymbol{I})$, $t = T$. Init buffer with empty sets: $\texttt{buffer}(t) \leftarrow \{\}$, $t = 1, 2, \cdots, T$.

**while** $t > 0$ **do**

    Compute the conditional mean $\boldsymbol{x}_{0|t} = \frac{\boldsymbol{x}_t - \sigma_t \epsilon_\theta(\boldsymbol{x}_t, t)}{\alpha_t}$, and estimate the verifier score $f(\boldsymbol{x}_{0|t})$.

    Add the score-sample pair to the buffer: $\texttt{buffer}(t).\texttt{add}\left(f(\boldsymbol{x}_{0|t}) : \boldsymbol{x}_t\right)$

    **if** $f(\boldsymbol{x}_{0|t}) < \delta_t$ and budget $B > 0$ **then**

        Backtrack: $t_{\text{next}} \leftarrow \min(t + \Delta_T, T)$, $\boldsymbol{x}_{t_{\text{next}}} \sim q(\boldsymbol{x}_{t_{\text{next}}}|\boldsymbol{x}_t)$ with $q$ in Eq. 1

        Decrease the budget: $B \leftarrow B - 1$

    **else**

        **if** $B = 0$ **then**

            Pop the best sample from buffer: $\boldsymbol{x}_t \leftarrow \texttt{buffer}(t).\texttt{max}$   ▷ *select the best sample from past explorations*

        **end if**

        Sample posterior: $t_{\text{next}} \leftarrow t - 1$, $\boldsymbol{x}_{t_{\text{next}}} \sim \tilde{p}_\theta(\boldsymbol{x}_{t-1}|\boldsymbol{x}_t)$

    **end if**

    $t \leftarrow t_{\text{next}}$, $\boldsymbol{x}_t \leftarrow \boldsymbol{x}_{\text{next}}$

**end while**

Return $\boldsymbol{x}_0$

---

# E  EXPERIMENT DETAILS

In this section we provide the details of experimental setup and implementation for all our experiments. We run our experiments on clusters with Nvidia A100 and RTX 4090 GPUs, with over 1000 GPU hours used.

## E.1  ABLATION OF BFS DESIGN SPACE

We directly adopt the official code base of FK-steering (Singhal et al., 2025a) and use the samping methods provided in the code base of DAS (Kim et al., 2025). We use the ImageReward prompts as in Singhal et al. (2025a) and report the average and standard deviation over 4 independent trials. For the temperature and resampling interval, we directly follow the implementation of FK-steering. For TreeG (Guo et al., 2025) we use a fixed branch out size of 2. For Dsearch (Li et al., 2025b) we set the ratio $\frac{b(T)}{b(0)} = 4$ as in the original paper, and for SoP (Ma et al., 2025) we directly adopt the noise levels of the original paper, and use Paths-2 for $N = 4$ and Paths-4 for $N = 8$. For the FLUX model, we use the stochastic sampler from Wang & Yu (2025). The detailed hyper-parameters are below in Table. 6:

Table 6: BFS implementation parameters

| | |
|---|---|
| temperature | 10 |
| resampling interval | [20,40,80] |
| sampling steps | 100 |
| num seeds | 4 |

### E.1.1  ADDITIONAL RESULTS FOR THE BFS EXPERIMENTS

In this section, we provide additional results of the BFS experiments.

In the original implementation of DAS (Kim et al., 2025), the proposal transition kernel is biased with the verifier gradient $\tau = \mathcal{N}\left(\boldsymbol{x}_{t-1}; \mu_\theta(\boldsymbol{x}_t) + \lambda\nabla_{\boldsymbol{x}_t}\log f(\boldsymbol{x}_{0|t}), \sigma_t\right)$, and include the log ratio in the resampling weights: $w_t^i = \log p_\theta(\boldsymbol{x}_{t-1}^i|\boldsymbol{x}_t^i) - \log\tau(\boldsymbol{x}_{t-1}^i|\boldsymbol{x}_t^i) + \lambda\log f(\boldsymbol{x}_t^i) - \lambda\log f(\boldsymbol{x}_{t-1}^i)$.

In Table. 2, we disable all the gradients in global search methods for a fair comparison. Here we provide the results using the original implementation with gradients (DAS-grad) in Table. 7. We tune the KL coefficient so that the relative magnitude of the reward divided by the KL coefficient remains consistent with the experiments in the original paper (Kim et al., 2025). We also evaluate our BFS method with the gradient-biased kernel (BFS-grad), where we use the KL coefficient from the DAS-grad method as the guidance strength, ensuring an identical transition kernel. From the results, we observe that both ImageReward and HPS scores benefit from the additional gradient guidance, with our BFS design enabling more efficient global exploration.

To validate the robustness of inference-scaling, we further evaluate the generated samples with Human Preference Score (Wu et al., 2023b), and provide visualizations in Fig. 6.

| | | BoN | | DAS (w/o grad) | | DAS-grad | | BFS (w/o grad) | | BFS-grad | |
|---|---|---|---|---|---|---|---|---|---|---|---|
| Model | N | ImageReward | HPS | ImageReward | HPS | ImageReward | HPS | ImageReward | HPS | ImageReward | HPS |
| SD v1.5 | 4 | $0.702 \pm 0.057$ | $0.264 \pm 0.001$ | $0.878 \pm 0.028$ | $0.265 \pm 0.001$ | $1.084 \pm 0.052$ | $0.269 \pm 0.002$ | $0.882 \pm 0.029$ | $0.265 \pm 0.001$ | $\mathbf{1.191 \pm 0.041}$ | $\mathbf{0.271 \pm 0.001}$ |
| SD v1.5 | 8 | $0.896 \pm 0.031$ | $0.267 \pm 0.001$ | $1.052 \pm 0.033$ | $0.268 \pm 0.001$ | $1.197 \pm 0.053$ | $0.271 \pm 0.001$ | $1.087 \pm 0.031$ | $0.268 \pm 0.001$ | $\mathbf{1.302 \pm 0.043}$ | $\mathbf{0.273 \pm 0.002}$ |
| SD XL | 4 | $1.085 \pm 0.013$ | $0.273 \pm 0.001$ | $1.181 \pm 0.023$ | $0.275 \pm 0.001$ | $1.272 \pm 0.047$ | $0.277 \pm 0.002$ | $1.194 \pm 0.024$ | $0.275 \pm 0.001$ | $\mathbf{1.413 \pm 0.038}$ | $\mathbf{0.278 \pm 0.001}$ |
| SD XL | 8 | $1.198 \pm 0.021$ | $0.277 \pm 0.001$ | $1.265 \pm 0.019$ | $0.278 \pm 0.001$ | $1.485 \pm 0.046$ | $0.279 \pm 0.001$ | $1.291 \pm 0.018$ | $0.279 \pm 0.001$ | $\mathbf{1.532 \pm 0.035}$ | $\mathbf{0.281 \pm 0.001}$ |

Table 7: Additional BFS experiment results

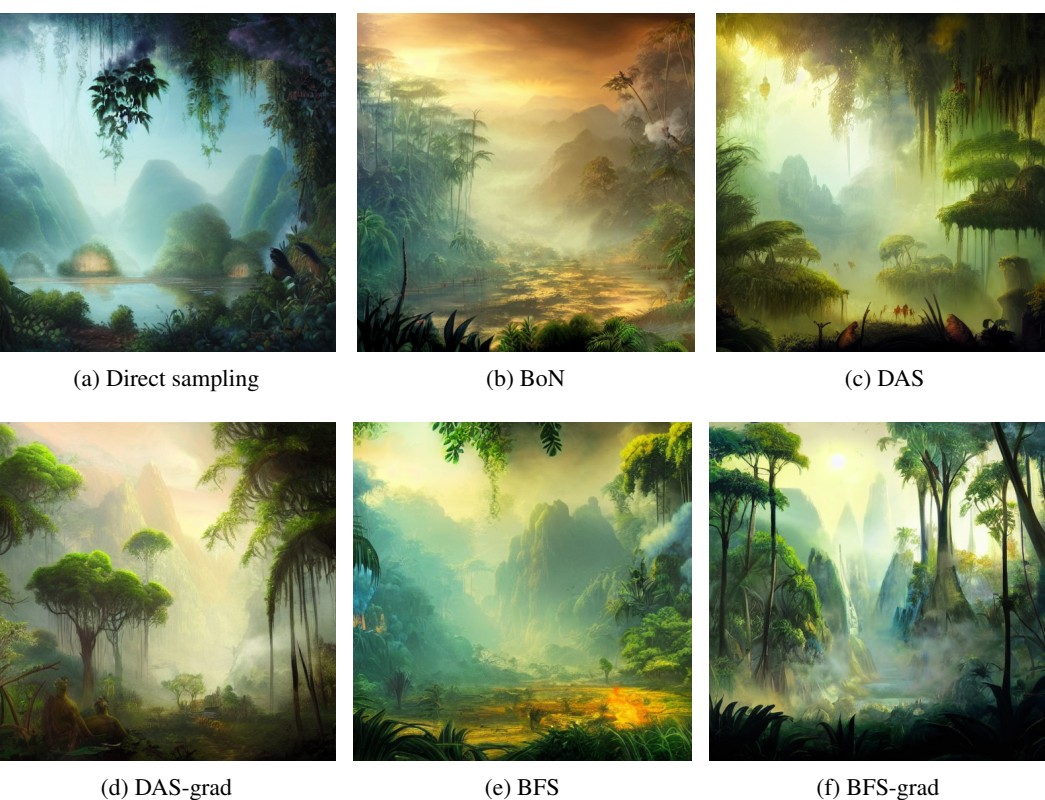

(a) Direct sampling    (b) BoN    (c) DAS

(d) DAS-grad    (e) BFS    (f) BFS-grad

Figure 6: Visualizations for SD v1.5 generated samples, where direct sampling denotes sampling without inference-scaling, and the number of particles is N=4 for other methods. Prompt: a beautiful painting of the jungle in the morning with lots of smoke, fantasy art, matte painting.

## E.2 TEXT-TO-IMAGE COMPOSITIONAL GENERATION WITH DFS

We use the SSD-1B model[7] which is distilled from SDXL, and we use the default sampling configuration with 50 steps of DDIM sampler. For DFS and BFS, we evaluate at time steps $\{25, 35, 45\}$ and set the backtrack depth $\Delta_T = 25$. For BFS we additionally sweep the temperature in range $\{0.5, 1, 2, 4, 8\}$ and report the best performance.

We provide example visualizations in Fig. 7.

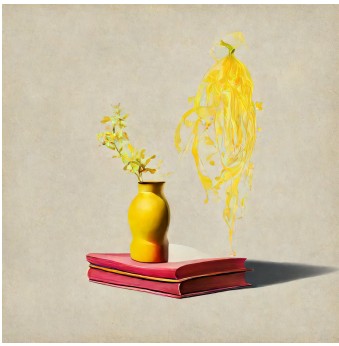
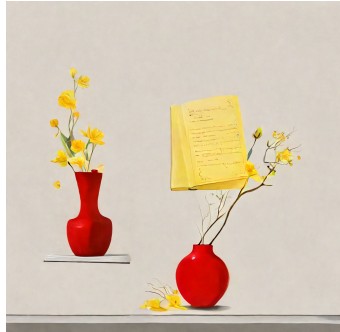

(a) w/o inference scaling         (b) w/ inference scaling

(c) Example from color dataset with prompt: a yellow book and a red vase

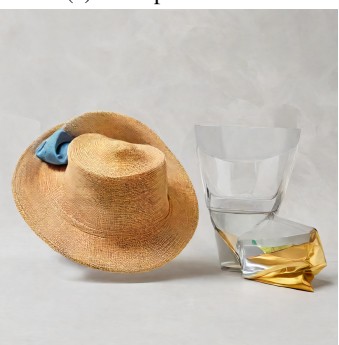
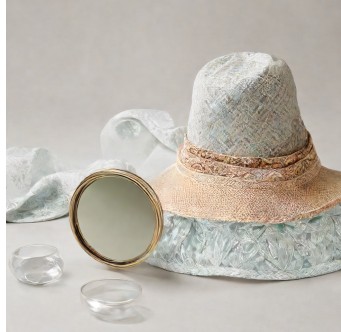

(d) w/o inference scaling         (e) w/ inference scaling

(f) Example from texture dataset with prompt: a fabric hat and a glass mirror

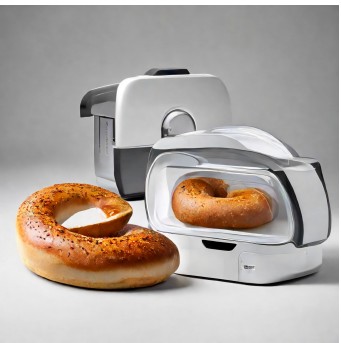
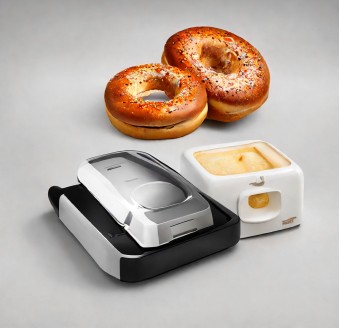

(g) w/o inference scaling         (h) w/ inference scaling

(i) Example from shape dataset with prompt: a round bagel and a square toaster

Figure 7: **Example of inference-time scaling with CompBench** Above shows the example prompts from CompBench benchmark and images before and after inference scaling

**Comparison with SoP**. Compared with SoP (Ma et al., 2025), our DFS adopts an adaptive backtracking schedule, which enables efficient quality-dependent exploration and full restarts to escape

[7]https://huggingface.co/segmind/SSD-1B

low quality modes. We provide the additional experiment results in Fig. 8. For SoP, we follow the original backtracking schedule, and the uses Paths-2 for low compute and Paths-4 for scaling up more compute, following the insights of Ma et al. (2025). We use $\delta = 0.7$ for DFS.

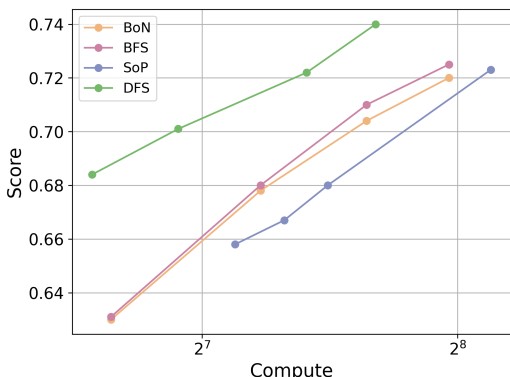

Figure 8: Comparison of DFS with SoP

### E.3 LONG HORIZON MAZE PLANNING

**Maze environment**. For all our maze experiments we use the OGBench PointMaze environment (Park et al., 2024). We created our maze layout using the same protocol of Figure 5 in Marcucci et al. (2023)[8], but with a smaller size of 20x20 cells. Dataset is collected following the protocal in OGBench (Park et al., 2024). We evaluate the model on the default task 1 of OGBench (Park et al., 2024), which is navigating from bottom left to top right. Empirically we discover that the diffusion model can perform well on short-horizon tasks without extra inference compute, but struggles heavily in the long horizon tasks.

**Model Training**. We train the model following diffuser (Janner et al., 2022), where we use a temporal U-Net to denoise the trajectory

$$\boldsymbol{\tau} = \left[ \begin{array}{cccc} \boldsymbol{s}_1 & \boldsymbol{s}_2 & \cdots & \boldsymbol{s}_H \\ \boldsymbol{a}_1 & \boldsymbol{a}_2 & \cdots & \boldsymbol{a}_H \end{array} \right] .$$

Since our objective start and goal is more distant than trajectories in dataset, we sample at longer horizons than training, which is enabled by the temporal U-Net architecture. We train the model for 1.2M steps using the same configuration as Janner et al. (2022). Model details are below in Table. 8.

Table 8: Hyper-parameters for maze model

| hidden dimension multipliers | [1,4,8,16] |
|---|---|
| training horizon | 2000 |
| sampling horizon | 2800 |
| training steps | 1.2M |
| batch size | 64 |
| learning rate | 1e-4 |

**Inference**. We found that the model performance saturates with 16 denoising steps, which we use for all our experiments. For all the data points we report the average success rate with over 40 samples.

For verifier design, we use the ground-truth maze layout, and calculate the violation of each point in the trajectory using the position coordinates. Specifically, if a point $(x, y)$ is inside a maze wall box with center $(c_x, c_y)$ and half-width $d$, then the point loss can be calculated as the minimum distance from the point to box walls:

$$L(x, y) = \min\left(x - (c_x - d), (c_x + d) - x, y - (c_y - d), (c_y + d) - y\right) .$$

---

[8]https://github.com/mpetersen94/gcs/blob/main/models/maze.py

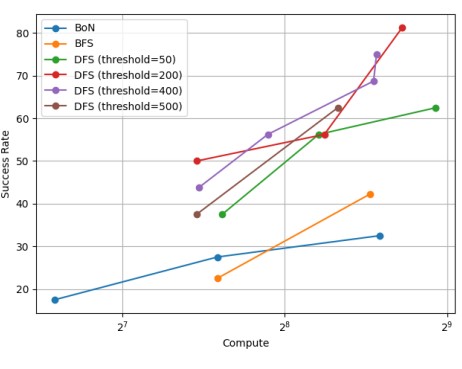 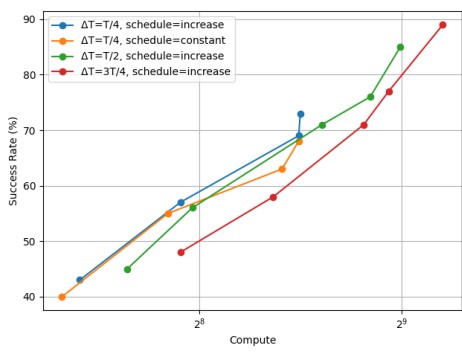

(a) Ablation of threshold $\delta$

(b) Ablation of backtrack schedule

Figure 9: Ablation of DFS hyperparameters

and the total verifier score is computed as:

$$f(\boldsymbol{\tau}) = \exp\left(-\sum_{i=1}^{H} L(x_i, y_i)^2\right).$$

So if all the points are free of violation in the trajectory, then $f(\boldsymbol{\tau}) = 1$. We point out that this does not indicate a successful plan as the connection between consecutive points $(x_i, y_i) \rightarrow (x_{i+1}, y_{i+1})$ may violate the maze layout, and using only the verifier function can not generate a successful plan.

For local search, we search the hyper-parameters $\bar{\rho}$ and $\bar{\mu}$ in Sec. C.5 with $\bar{\gamma} = 0$. For global search with BFS we evaluate at steps $\{12, 8, 4\}$, and for DFS we evaluate at $\{12, 11, \cdots, 1\}$ with backtracking depth $\Delta_T = 4$. We also observe that increasing backtracking depth to 12 and evaluate at smaller time steps $\{4, 3, 2, 1\}$ helps to scale up the performance with more compute. The hyperparameter search results are below in Table.9 and Fig. 9

| N | $\tau = 0.2$ | $\tau = 0.005$ | $\tau = 0.1$ |
|---|---|---|---|
| 2 | $27.5 \pm 4.3$ | $32.5 \pm 1.1$ | $31.2 \pm 4.2$ |
| 4 | $42.5 \pm 5.2$ | $48.1 \pm 1.1$ | $45.5 \pm 2.3$ |
| 8 | $67.6 \pm 1.1$ | $71.2 \pm 2.2$ | $70.1 \pm 1.1$ |

Table 9: Hyperparameter search for temperature $\tau$ in PointMaze BFS

### E.4 OFFLINE RL

**Background**. Diffusion policy (Chi et al., 2023) is widely used for action generation in robot foundation models (Black et al., 2024; Liu et al., 2024). At inference time, policies can be guided by human trajectory constraints (Wang et al., 2022) or LLM-based value functions (Nakamoto et al., 2024). Exact diffusion sampling requires training a noise-dependent energy function (Lu et al., 2023), but this can degrade pretrained knowledge and demands additional data—often impractical in data-scarce robotic settings. In contrast, inference-scaling provides a more flexible approach, allowing seamless composition of pretrained diffusion policies with Q-functions without retraining.

**Setup**. We follow the setup in Lu et al. (2023), and we directly use their pretrained diffusion model and Q-function, omitting the time-dependent energy function. The diffusion model was trained to generate action $\boldsymbol{a}$ given state $\boldsymbol{s}$, and we sample with 15 steps of DDIM.

For hyper-parameter search, we disable the implicit dynamics and set $\bar{\gamma} = 0$, and use the 'increase' schedule for $\boldsymbol{\rho}$ and $\boldsymbol{\mu}$. For strength parameters $\bar{\rho}$ and $\bar{\mu}$, we first search for the right magnitude. Then, we also follow Lu et al. (2023) and search with step size [1,2,3,5,8,10] within the magnitude. Same as Lu et al. (2023), we use 5 different seeds with 10 samples per seed for each task. To avoid over fitting, we use different seeds for parameter search and evaluation. We report the hyper-parameters and the performance within the parameter-searching dataset and evaluation dataset.

For global search, we use 4 particles for Medium-Expert and Medium datasets, and 2 particles for Medium-Replay datasets. Since the number of particles are small, we do not carry out BFS or DFS methods and simply use Best-of-N. We point out that the number of particles we use are much smaller than the 50 particles in Wang et al. (2022) and the 32 particles in Chen et al. (2023), highlighting the effectiveness of local search.

**Baseline**. We compare our method to a variety of baselines, including traditional state-of-the-art methods IQL (Kostrikov et al., 2021) and diffusion-based policies such as diffuser (Janner et al., 2022), decision-diffuser (DD) (Ajay et al., 2022), Diffusion-QL (D-QL) (Wang et al., 2022), SfBC (Chen et al., 2023) and QGPO (Lu et al., 2023). We directly take the numbers from Lu et al. (2023).

Among the baseline diffusion-based methods, both Diffuser (Janner et al., 2022) and QGPO (Lu et al., 2023) requires training a noise-dependent guidance function, and D-QL (Wang et al., 2022) requires updating the diffusion model during training using the Q-function iteratively, which needs to back-propagate through the diffusion sampling chain, introducing high computation and memory overheads. DD (Ajay et al., 2022) uses classifier-free guidance (Ho & Salimans, 2022) to generate high-return trajectories that requires training a return-conditional model on labeled datasets, which can be expensive to obtain in robotics where only demonstration data is available (Liu et al., 2024).

For our reproduced baselines, TFG (Ye et al., 2024) is allowed up to 8 recurrence steps and DAS (Kim et al., 2025) up to 16 particles, resulting in a hyperparameter space and computational cost approximately twice that of our method. We sweep across all configurations for the baseline methods and report the best performance. For fair comparison we evaluate our method on different seeds used for hyperparameter search, with the results shown in Table 10.

| Dataset | Environment | particles | $N_{recur}$ | $N_{iter}$ | $\bar{\rho}$ | $\bar{\mu}$ | Eval set | Search set |
|---|---|---|---|---|---|---|---|---|
| Medium-Expert | HalfCheetah | 4 | 1 | 1 | 0.008 | 0.02 | $93.9 \pm 0.3$ | $\mathbf{94.3 \pm 0.5}$ |
| Medium-Expert | Hopper | 4 | 1 | 4 | 0.001 | 0.00 | $104.4 \pm 3.1$ | $\mathbf{109.4 \pm 5.2}$ |
| Medium-Expert | Walker2d | 4 | 1 | 1 | 0.005 | 0.10 | $\mathbf{111.4 \pm 0.1}$ | $111.4 \pm 0.2$ |
| Medium | HalfCheetah | 4 | 1 | 4 | 0.003 | 0.05 | $\mathbf{54.8 \pm 0.1}$ | $54.8 \pm 0.2$ |
| Medium | Hopper | 4 | 4 | 4 | 0.003 | 0.02 | $99.5 \pm 1.7$ | $\mathbf{100.1 \pm 0.1}$ |
| Medium | Walker2d | 4 | 1 | 6 | 0.003 | 0.08 | $\mathbf{86.5 \pm 0.2}$ | $85.2 \pm 3.2$ |
| Medium-Replay | HalfCheetah | 2 | 1 | 6 | 0.005 | 0.03 | $47.8 \pm 0.4$ | $\mathbf{48.4 \pm 0.1}$ |
| Medium-Replay | Hopper | 2 | 1 | 1 | 0.003 | 0.20 | $97.4 \pm 4.0$ | $\mathbf{100.4 \pm 2.2}$ |
| Medium-Replay | Walker2d | 2 | 2 | 4 | 0.003 | 0.03 | $79.3 \pm 9.7$ | $\mathbf{83.2 \pm 2.8}$ |
| **Average** | | | | | | | 86.1 | **87.5** |

Table 10: Hyper-parameters on D4RL locomotion tasks with test-time scaling. We report the performance on hyper-parameter search dataset and the evaluation dataset, highlighting the best number.

### E.4.1    OFFLINE POLICY DISTILLATION

**Setup**. We adopt the `Medium` dataset from D4RL (Fu et al., 2020) and the corresponding pretrained models from Lu et al. (2023) as in the previous section. For the dataset, we replace the action of each state with the sample generated with test-time search (TTS), using the hyper-parameters in Table. 10. We then finetune the models on the modified dataset with early-stopping. For evaluation of finetuned models, we use 5 different seeds different from the seed used during training, with 10 samples per seed. During evaluation, we sample from the finetuned diffusion model directly, without the presence of any verifier. Detailed training hyper-parameters are below in Table. 11.

| | |
|---|---|
| epochs | 5 |
| learning rate | 1e-4 |
| batch size | 16384 |
| eval every epoch | 1 |
| sampling steps | 15 |

Table 11: Hyper-parameters for policy distillation

For the Baseline DPPO (Ren et al., 2024), we directly adopt the numbers reported in their paper, which uses the same Medium dataset and models as ours.

The training curves for policy finetuning are shown in Fig. 10. In both Hopper and Walker2d environments, performance converges rapidly, within a single epoch over the dataset. Notably, our method is significantly faster than online finetuning approaches, as it does not require online data collection. Training one epoch on the full Medium dataset takes only about 4 minutes on a single Nvidia RTX 4090 GPU, including the time for data generation via inference-scaling sampling, whereas DPPO requires several hours to reach convergence.

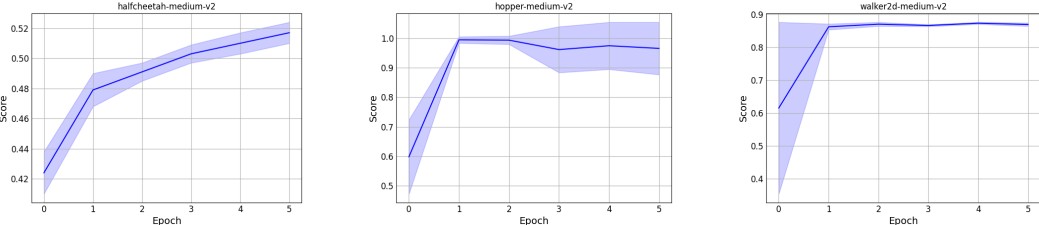

Figure 10: Training curves for policy distillation

### E.5 Mitigating Reward Hacking With Double Verifier

**Experimental setup**. We evaluate the proposed double-verifier on the challenging conditional ImageNet generation task, generating target-class samples from an unconditional model guided by a pretrained classifier. Specifically, we use two independent classifiers as verifiers[9][10] for global and local search. We report the Fréchet Inception Distance (FID) computed on 256 generated samples against the corresponding ImageNet class, and measure class accuracy using a separate classifier[11]. Since we only apply the global verifier sparsely, double-verfier introduces negligible computational costs. For local search we search the hyper-parameters with step size [1,2,4,8,10], and for global search with BFS, we search the temperature in [1.0,2.0,4.0]. For MSP score, we adopt a different imagenet classifier [12].

For the MSP score (Hendrycks & Gimpel, 2016), it is defined as:

$$\mathrm{MSP}(\boldsymbol{x}) = \max_c p(c|\boldsymbol{x})$$

Thus, higher MSP score indicates higher confidence that the image belongs to one of the classes from the dataset, which is less OOD.

**Visualizations**. Here we provide some visualizations of adversarial reward hacking and the effectiveness of our double-verifier approach. As shown in Fig. 11, we can see the adversarial patches that exploits the weakness of a single verifier, making the verifier to predict the images as belonging to the wrong class.

## F Wall-Clock Time Analysis

We conduct a wall-clock time analysis of our methods on the class-guided ImageNet generation task, measuring the average per-sample runtime when generating 10 samples under similar GPU utilization on a single RTX 4090. In DAS Kim et al. (2025) they only compute the verifier gradient while in our local search we additionally incorporates the recurrence in the Langevin MCMC step. We observe that wall-clock time is primarily dominated by the number of NFEs. Local search and DAS (Kim et al., 2025) is slower due to the additional gradient computations, while DFS is slightly slower because it generates particles sequentially, preventing full utilization of GPU batch parallelism.

---

[9] https://huggingface.co/google/vit-base-patch16-224
[10] https://huggingface.co/google/vit-base-patch16-384
[11] https://huggingface.co/facebook/deit-small-patch16-224
[12] https://huggingface.co/facebook/deit-base-patch16-224

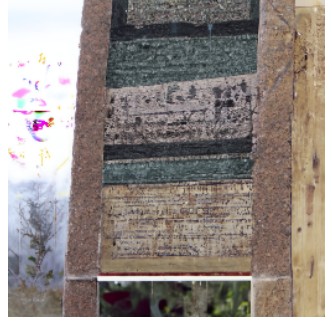 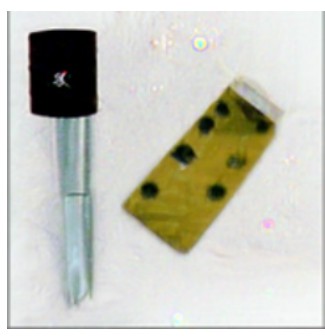 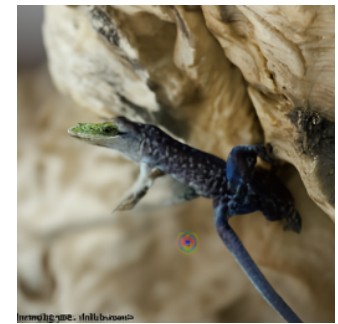

(a) Adversarial example with 0.96 class validity predicted by the local search verifier

(b) Adversarial example with 0.95 class validity predicted by the local search verifier

(c) Adversarial example with 0.98 class validity predicted by the local search verifier

Figure 11: **Examples of adversarial samples generated by gradient over-optimization** Here we provide the adversarial samples generated by gradient guidance on class 222 (kuvasz). Although the samples have high probability of belonging to the target class as predicted by the verifier, they are caused by adversarial reward hacking.

| Method | Settings | NFEs of diffusion model | Seconds |
|---|---|---|---|
| Direct sampling | N=1 | 100 | 14 |
| DAS | N=1, 1 gradient step | 100 | 26 |
| Local search | N=1, 1 additional local search step | 200 | 40 |
| Local search | N=1, 2 additional local search steps | 300 | 66 |
| BoN | N=4 | 400 | 55 |
| BFS | N=4 | 400 | 56 |
| DAS | N=4, 1 gradient step | 400 | 106 |
| DFS | budget=4 | 210 | 29 |
| BoN | N=4, 2 additional local search steps | 1200 | 260 |
| BFS | N=4, 2 additional local search steps | 1200 | 261 |
| DFS | budget=4, 2 additional local search steps | 620 | 136 |

Table 12: Wall-clock time analysis

## G    LLM USAGE STATEMENT

In this paper, LLMs are used exclusively for grammar checks, with no changes made to any technical components.

