# OpenReview forum: "Inference-time scaling of diffusion models through classical search"
_ICLR.cc/2026/Conference — ICLR 2026 Poster_

### Official Review · Reviewer_Vc1j · 2025-10-28

**Soundness:** 3
**Presentation:** 4
**Contribution:** 3
**Rating:** 8
**Confidence:** 4

**Summary:**

This paper proposes a unified framework for inference-time scaling of diffusion models by leveraging principles from classical search. The authors formulate the inference process as a combination of global and local search. For global search, they employ tree-search algorithms such as Breadth-First Search (BFS) and Depth-First Search (DFS) to efficiently identify high-quality modes in the generative space. To further refine sample quality beyond the base model's distribution, they introduce a theoretically-grounded local search based on annealed Langevin MCMC. The proposed method demonstrates superior or comparable performance across a range of challenging domains, including long-horizon planning, offline reinforcement learning, and image generation. The paper is supported by comprehensive empirical analyses of its design choices and a theoretical convergence analysis in the appendix.

**Strengths:**

- **Clarity and Thoroughness of the Proposed Method:** The paper provides a detailed and well-structured discussion of the proposed methods, both in the main text and the appendix. This thoroughness contributes to a clear and concrete presentation of their framework.

- **Comprehensive Empirical Analysis:** The authors systematically present and analyze the empirical results for each component of their method. For example, they methodically ablate the design choices for BFS (resampling, scoring, and tempering) and subsequently demonstrate the efficiency gains achieved through the introduction of DFS and local search.

- **Unified Framework for Inference-Time Scaling:** The paper successfully presents a unified framework that integrates global and local search strategies. This framework cohesively combines key functionalities essential for controlled generation, such as branching, exploration, backtracking, and optimization beyond the pretrained distribution.

**Weaknesses:**

- **Large Number of Hyperparameters:** While powerful, the proposed integrated framework introduces a significant number of hyperparameters. These include the BFS tempering parameter () and number of particles, the DFS backtracking step size (​) and score threshold (), and the local search steps (​), among others. As indicated by the experimental results, performance is sensitive to the tuning of these parameters. The authors acknowledge this as a limitation in the "Limitations and Conclusion" section and suggest automatic tuning algorithms as future work. I agree with their assessment that this is a key area for improvement.

- **Fixed Backtracking Step Size $\Delta_T$:** In the proposed DFS algorithm, when a sample's estimated score falls below a predefined threshold $\delta$, the algorithm backtracks by adding a fixed amount of noise determined by the hyperparameter $\Delta_T$​. However, the optimal amount of noise to inject may vary depending on the specific instance. In some cases, a small perturbation might be sufficient to escape a low-quality region, while in others, a near-complete restart from a higher noise level might be necessary. The current method lacks the flexibility to handle this adaptively. In contrast, other contemporary works have explored more dynamic approaches. For instance, Adaptive Bi-directional Cyclic Diffusion (ABCD) [1] addresses this by propagating particles across different noise levels to balance exploration and exploitation. Similarly, Monte Carlo Tree Diffusion (MCTD) [2] uses a selection policy inspired by MCTS to navigate the search space more dynamically.

- **The Statement about the Use of Large Language Model:** The authors doesn’t state explicitly how they used LLM in their submission, which is guided in https://iclr.cc/Conferences/2026/AuthorGuide.

- **(Minor) Presentation Issues:**
    - In line 200 on page 4, there appears to be a typo in the transition kernel notation: $xt-1$ should likely be written in LaTeX as $x_{t-1}$.

[1] Lee, Gyubin, et al. "Adaptive Cyclic Diffusion for Inference Scaling." _arXiv preprint arXiv:2505.14036_ (2025).

[2] Yoon, Jaesik, et al. "Monte carlo tree diffusion for system 2 planning." _arXiv preprint arXiv:2502.07202_ (2025).

**Questions:**

- Could you please provide an empirical ablation study on the backtracking hyperparameter $\Delta_T$​? Understanding its sensitivity would be valuable for practitioners seeking to apply your method.

- Regarding the scoring functions in BFS, I understand that 'Difference' and 'Max' scoring account for the trajectory's history. Could you elaborate on the necessity of this? What are the specific advantages of scoring based on the trajectory history compared to only using the score from the current timestep?

- As mentioned in the "Limitations and Conclusion" section, using evolutionary algorithms for hyperparameter tuning is an interesting direction. However, an alternative direction for future work could be to further unify the three search strategies (DFS, BFS, and local search) to be controllable by a smaller, more concise set of core hyperparameters. I would be interested to hear the authors' thoughts on the feasibility and potential of such a direction.

---

> ### Author Response · Authors · 2025-11-18
> **Reply to Reviewer Vc1j**
>
> We thank the reviewer for the valuable and constructive review, and we have revised accordingly, including a LLM usage statement.
>
> > Large Number of Hyperparameters
>
> As shown in the DFS ablations Figure 3 (left), Figure 9 (Page 31) and BFS abations in Table 9 (Page 31), the performance is robust across different hyperparameter choices.
>
> > Fixed Backtracking Step Size
>
> Since we evaluate particles at different time steps, the same backtracking depth can lead to restarts from different regions. For example, if a particle is of very low quality, it may be rejected at an earlier step, thereby triggering a full restart. Conversely, if a particle passes earlier thresholds and is only rejected at a low noise level, a global restart is unnecessary, and we can restart from a lower noise level.
>
> In contrast, the concurrent work [1] allows particles to move across different noise levels to balance exploration and exploitation, but it lacks the flexibility to reject particles at early time steps. The MCTS framework in [2] is designed for trajectory generation and gradually denoises trajectories by handling different segments at different noise levels. This is fundamentally different from our setting, where we denoise the entire trajectory jointly rather than segment by segment.
>
>
> > Empirical ablation study on the backtracking hyperparameter $\Delta_T$?
>
> We have conducted an ablation of DFS in the Maze task, presented in Figure 9 (Page 31) [https://postimg.cc/qNPnHdzk](https://postimg.cc/qNPnHdzk). The results show that DFS remains robust across $\Delta_T$ choices, and larger $\Delta_T$ helps to scale up performance given more compute. We suggest practitioners to start with relatively small backtrack depth such as $\Delta_T=T/4$ and increase $\Delta_T$ with more compute budget.
>
> > Elaboration about scoring function
>
> In DAS [3] and other SMC-based methods, the resampling probability is determined by the reward difference between consecutive steps, which provides a theoretical asymptotic convergence guarantee for sampling from the target distribution. Empirically, this design prioritizes samples that continue to improve during the denoising process. However, as noted by FK-Steering [4], this approach can be unfair to sampling paths that saturate early, and the authors therefore propose using Max scoring instead. Within our BFS framework, we experimented with all scoring options, and generally found that they have only marginal impact on the final results.
>
> > Discussion on unified and concise parameter space
>
> For global search methods, it is possible to select initial hyperparameters based on reward statistics. For example, in BFS we can set the temperature by controlling the variance of the temperature-weighted rewards, which transfers well across different tasks. Similarly, in DFS, the backtracking threshold can be chosen as a specific percentile of the reward distribution to control the probability of triggering backtracks. In general, these parameters reflect exploration–exploitation trade-offs, and more principled tuning strategies could be developed. For local search methods, it would also be interesting to investigate whether a more unified parameterization can be designed.
>
> -----
> [1] Lee, Gyubin, et al. "Adaptive Cyclic Diffusion for Inference Scaling." arXiv preprint arXiv:2505.14036 (2025).
>
> [2] Yoon, Jaesik, et al. "Monte carlo tree diffusion for system 2 planning." arXiv preprint arXiv:2502.07202 (2025).
>
> [3] Kim, Sunwoo, Minkyu Kim, and Dongmin Park. "Test-time alignment of diffusion models without reward over-optimization." arXiv preprint arXiv:2501.05803 (2025).
>
> [4] Singhal, Raghav, et al. "A general framework for inference-time scaling and steering of diffusion models." arXiv preprint arXiv:2501.06848 (2025).

---

> > ### Comment · Reviewer_Vc1j · 2025-11-24
> >
> > Thank you for the detailed rebuttal and additional experiments. The ablation study on the score threshold and backtrack step size has fully answered my questions about performance robustness. (Please note that the subfigure captions in Figure 9 look switched).
> >
> > I also find the response regarding the backtrack step size valid, which clears up my concerns about adaptive backtracking. Finally, I appreciate the clarification regarding the score function.

---

> > > ### Author Response · Authors · 2025-11-24
> > >
> > > Thanks for your feedback! We have fixed the subfigure captions in Figure 9. We’re happy to discuss any further questions you may have.

---

### Official Review · Reviewer_JCZg · 2025-11-01

**Soundness:** 2
**Presentation:** 3
**Contribution:** 3
**Rating:** 6
**Confidence:** 4

**Summary:**

This paper focuses on inference-time control of diffusion models to generate samples that maximize given objectives. It combines global search via breadth-first and depth-first search and local search via annealed Langevin MCMC. The method is applied across planning, offline reinforcement learning, and image generation to demonstrate improved performance and efficiency over baselines.

**Strengths:**

1. It explores various design choices in inference-time scaling of diffusion models across global and local search and proposes a unified framework that jointly scales both global and local search.
2. It proposes a method using depth-first search (DFS), which can adaptively allocate compute.
3. Decomposed ablation on global search (Section 5.2) and local search (Section 5.3.1) shows the effectiveness and necessity of both methods.
4. Experiment demonstrates that distilling TTS samples can be a powerful off-policy RL method. (I believe [1] has already done this, though they didn't use the term 'inference-time scaling', but referred to it as global and local optimization, so being the 'first' work seems inaccurate.)

[1] MARK, Max Sobol, et al. Policy agnostic rl: Offline rl and online rl fine-tuning of any class and backbone. arXiv preprint arXiv:2412.06685, 2024.

**Weaknesses:**

1. Usage of Langevin MCMC prevents applying local search for non-differentiable rewards or naturally expanding the method to discrete diffusions. There are other MCMC alternatives, such as Metropolis-Hastings variants (e.g., predictor-corrector), but the choice of Langevin MCMC isn't justified sufficiently.
2. By reporting only a single reward value in image experiments, it's hard to tell whether there is severe reward hacking and the method is generating broken images. For instance, excessive guidance can achieve high reward but generate a broken image that is off the data manifold. Please provide additional metrics (e.g., additional preference score or FID) or qualitative results to show the proposed method is searching for high-reward samples on the data manifold.

**Questions:**

1. The claim that local search via Langevin MCMC can 'go beyond base model', which repeatedly appears in the paper (Section 1 line 51-52, line 80-81, Section 4.2), may be misleading. Langevin MCMC still generates samples from the compositional distribution $p_0(x)f(x)^\lambda$ where the support is a subset of the support of $p_0$. Thus, similar to global search, it's searching for high-reward samples in the support of the base model. Even if it can explore very low-probability (relative to the base model) regions, there seems to be no direct evidence in the paper.
2. SVDD, DAS, and FK-Steering all fall under the umbrella of SMC. Categorizing them as BFS seems like simple reframing. Also, when resampling, does it consider log likelihood with respect to the base model, which is the case for DAS and FK-Steering?
3. What is TTS in Section 5.3.2? Is it BFS + local search or DFS + local search?
4. What's the purpose of double-verifier experiments? Solving reward hacking is an important problem, but the experiments don't demonstrate the additional benefit of the proposed BFS. It's rather just introducing an additional technique for a separate problem.

---

> ### Author Response · Authors · 2025-11-18
> **Reply to Reviewer JCZg (1/2)**
>
> We thank the reviewers for their thoughtful and constructive feedback. Below, we provide additional discussion and results.
>
> > Relationship with Policy-agnostic RL
>
> We thank the reviewers for highlighting this important and relevant paper, and we have added a discussion of it in the revision. The key distinction is that our method relies entirely on a pretrained Q-function and performs no additional Q-updates, which allows for more flexible composition of pretrained verifiers and policies. In contrast, PA-RL requires iterative joint updates of both the actor and critic over hundreds of epochs. Our superior performance demonstrates the effectiveness of our inference-scaling framework compared with naïve action optimization.
>
> > The choice of Langevin MCMC isn't justified sufficiently
>
> We choose Langevin MCMC because we only have access to the score function $\nabla_x\log p_t(x)$ from the diffusion model, instead of the exact $p_t(x)$ or energy function $E_t(x)\propto p_t(x)$, which is required by the predictor-corrector methods such as Metropolis-Hastings MCMC or Metropolis adjusted Langevin MCMC.
>
> > It's hard to tell whether there is severe reward hacking and the method is generating broken images
>
> We have provided the HPS scores using the same ImageReward verifier in the BFS text-to-image experiments, along with the corresponding visualizations, in Section E.1.1 (Page 28). We include the relevant results below, where we can see that although we use the ImageReward verifier, HPS scores remain consistent.
>
>
> **HPS Scores of BFS experiment with ImageReward**
> | Model   | N   | BoN  | BFS |
> | ------- | --- | ---- | --- |
> | SD v1.5 | 4   |  0.264±0.001  |   **0.265±0.001**  |
> |     SD v1.5    |   8  | 0.267±0.001  |    **0.268±0.001** |
> |   SD XL      |   4  |   0.273±0.001   |  **0.275±0.001**   |
> | SD v1.5 | 8   | 0.277±0.001|  **0.279±0.001**   |
>
>
>
> In the DFS CompBench experiments, Figure 7 (Page 29) shows that our method improves the reward-relevant semantic content of generated images while maintaining their aesthetic quality. We share the reviewers’ concern about reward hacking. For the most vulnerable case, class-guided image generation, we therefore report both FID and class validity computed by an independent verifier to faithfully represent image quality and prevent reward hacking of the verifiers used in sampling.

---

> ### Author Response · Authors · 2025-11-18
> **Reply to Reviewer JCZg (2/2)**
>
> > The claim that local search via Langevin MCMC can 'go beyond base model' may be misleading, and there seems to be no direct evidence
>
> We have revised our rhetoric to more accurately describe the exploratory behavior of local search. In the offline RL experiments presented in Section 5.3.2 (Page 9), we show that without local search, pure global search methods such as SfBC significantly underperform local search, even with 16× more compute, especially on the Medium and Medium-Replay datasets where the base model has limited capabilities. These results indicate that local search can effectively sample from relatively low-probability regions of the base model’s manifold for high quality samples.
>
>
>
> >  Categorizing SMC-based methods as BFS seems like simple reframing. Also, when resampling, does it consider log likelihood with respect to the base model, which is the case for DAS and FK-Steering?
>
> Thank you for the suggestion. We have added additional discussion of SMC methods in Related works (Page 3) and Section B.3 (Page 16).
>
> For the global search experiments, because we disabled gradient computation in all methods for a fair comparison, the proposal kernel reduces to that of the base diffusion model. As a result, the log proposal probability and the log base model probability cancel out. This is consistent with the official implementation of FK-Steering which is gradient-free. We have added clarifications in Section 4.1.1 (Page 5). We also include the original version of DAS (DAS-grad), which uses the reward-model gradient to bias the transition kernel and include the transition probabilities during resampling. To compare the global search efficiency, we also evaluate our BFS with the same gradient biased transition kernel (BFS-grad). The additional results are reported in Section E.1.1 (Page 28). Relevant results are shown below. Empirically, we find gradient-enhanced proposal kernel improves performance, and our BFS demonstrates superior global search efficiency.
>
> **DAS results with original gradient enhanced implementation**
> | Model   | N   | BoN   | DAS   |  DAS-grad   | BFS-grad |
> | ------- | --- | ----- | ----- | --- | -------- |
> | SD v1.5 | 4   | 0.702±0.057 | 0.878±0.028 |  1.084±0.052   | **1.191±0.041**    |
> | SD v1.5 | 8   | 0.896±0.031 | 1.052±0.033 | 1.197±0.053    | **1.302±0.043**  |
> | SD XL   | 4   | 1.085±0.013 | 1.181±0.023 |  1.272±0.047   | **1.413±0.038**   |
> | SD XL   | 8   | 1.198±0.021 | 1.265±0.019 |  1.485±0.046   | **1.532±0.035**    |
>
>
>
>
>
> In our implementation of DAS in the local search experiments in Section 5.3 , we follow the original setup and retain the likelihood-difference term during scoring. Additionally, we further tune the KL coefficient with the same granularity as in our methods. This results in a stronger and carefully optimized baseline.
>
>
> > Questions about TTS
>
> As noted in lines 1675–1677 (Page 32), because our method uses only a small number of particles, we simply adopt BoN + local search. We have added this clarification to the main text.
>
>
> > Questions about the double-verifier experiment
>
> The double-verifier experiment introduces a method to reduce reward hacking in gradient-based optimization. This is particularly useful for our local-global joint scaling framework, where local search based on gradients is prone to vulnerabilities. We emphasize that this mechanism arises naturally from our unified framework, and is key to superior performance across domains such as class-guided image generation. Experiments with BFS and BoN show that the double-verifier strategy consistently improves performance, including for our strengthened BFS baseline.

---

### Official Review · Reviewer_TXqE · 2025-11-03

**Soundness:** 2
**Presentation:** 1
**Contribution:** 2
**Rating:** 2
**Confidence:** 4

**Summary:**

This paper studies inference-time scaling for diffusion models via: (i) a global search layer with a BF-style design space (tempering, scoring, resampling) meant to unify existing particle/tree methods; (ii) a DFS variant that backtracks to higher noise when a per-step verifier threshold is violated; and (iii) a local search layer using annealed Langevin MCMC. The authors also relate recurrent training-free guidance to Langevin MCMC in the small-step limit. Experiments span text-to-image (SD 1.5/SDXL/SSD-1B; CompBench), planning (PointMaze), and offline RL, plus a double-verifier trick for reward-hacking mitigation.

**Strengths:**

- The various design choices for the BFS method are clearly presented and the studies help understand which components are helpful.
- The threshold-based method is a useful modification for DFS that is practical, and experiments show it indeed scales up compute for harder problems as intended.
- The theoretical result bridge that guided recurrence approximates annealed Langevin MCMC in the small-step limit.
- The experiments span multiple domains including text-to-image generation, path generation in maze, and continuous control tasks for offline RL.

**Weaknesses:**

My main reservations concern the strength of the paper’s contribution and significance, the clarity of exposition, and a few statements that seem potentially misleading or inaccurate.

### Contribution

The global search methods BFS/DFS proposed in the paper largely seem to be an alternate perspective or modifications to existing methods, rather than fundamentally new algorithms

- The authors acknowledge that FK-steering/DAS are instantiations of BFS, but I do not see what new insights are provided by this new framing. The design choices for BFS in Section 4.1.1 are essentially ablation studies on existing techniques: FK-Steering [1] explored current/difference/max potential functions (and showed that max performs best), and DAS [2] showed that SSP sampling and tempering help improve performance. This is somewhat corroborated by Table 2, where BFS scores on all tasks are within one standard deviation of DAS scores.
- DFS modifies the fixed sequential noising/denoising procedure of SoP [3] with a threshold-based adaptive procedure, which is a useful modification, but it is again not a fundamentally new algorithmic contribution.

### Experiments

- **Comparison with SoP.** The paper tries to set apart DFS by stating that SoP uses “small noise injection for local exploration” as opposed to $\Delta_t \geq T/4$ for DFS. This is incorrect as SoP in fact uses $\Delta_t = 0.78\,T$. Please change this statement to accurately reflect prior work. In addition, despite the conceptual proximity, I find it a bit odd that there is no empirical comparison of DFS with SoP to understand the relative improvement of the proposed thresholding-based modifications.
- **DFS results.** The claim in Section 5.2 that DFS outperforms BFS/Best-of-N with “up to 2x less computational cost” is slight overclaiming based on the results in Figure 3 (left). The difficulty vs compute discussion in Figure 3 (right) does not show the resulting improvement from the additional compute. Please include baseline scores across the difficulty levels so we can compare the base model vs DFS results.
- **Hyperparameter burden.** The local search introduces step size and number of steps, DFS introduces threshold $\delta_t$ and backtracking depth. The paper has reasonable sweeps in its experiments, but introducing this many hyperparameters introduces a burden for practical adoption, as discussed briefly in the limitations. Specifically, regarding the threshold parameter of the DFS method which is introduced in this paper: (1) it is defined as a time-dependent threshold $\delta_t$, but it seems experiments use a constant value for all $t$? How should one vary this threshold with time? (2) This parameter generally seems hard to tune - given a new reward function with an arbitrary range, how shall a practitioner decide the suitable threshold value?
- **Insufficient experiment details in the main paper.** Many crucial experimental details that should be in the main paper are provided in the appendix, making the experiments section very hard to read.
    - Section 5.1 does not specify the prompt dataset used for evaluation. Appendix E.1 says the experiments used ImageReward prompts following the setting in [1], which actually used GenEval and DrawBench prompts for text-to-image. Please provide the correct citation to avoid confusion, and the prompt dataset should be specified in the main paper.
    - Section 5.3.2 does not specify the number of recurrent steps for local search that are used in the offline RL experiments. This information is not provided in the appendix as well.
    - Section 5.4 does not explain what the two different verifiers are used for ImageNet conditional generation; this information is only provided in Appendix E.5.

### Clarifications/corrections on statements

- Section 2 states that tree-search methods like [3,4] are special cases of the BFS framework. [3] seems closer to DFS as discussed later in the paper, and [4] does not map trivially to the BFS framework. An explanation to support this statement would be helpful.
- Section 3.2 discusses sampling from product distributions. However, the wording here is a bit misleading - it seems to suggest simply adding the score functions is enough to sample from product distributions. Also, authors define $\hat{q_t}(x_0)$ as a distribution on $x_0$ but the equation involves $\hat{q}_t(x_t)$ as a distribution over $x_t$. The appendix has a detailed explanation, but the wording here should be changed to avoid misunderstanding.
- The scoring functions for BFS use posterior mean predictions $x_{0|t}$ to get intermediate rewards at noisy states; asserting this “accounts for reward variation during sampling by considering not only the current reward but also its trajectory” is misleading since this prediction is specifically designed to *not* consider the sampling trajectories. Please qualify/modify this statement.
- The paper could benefit from discussing some relevant works: [5] uses an MCTS-style tree search, and [6] is a particle-based method that uses Gibbs sampling. Both methods outperform FK-steering and/or DAS by a significant margin.

*[1] Singhal, Raghav, et al. "A General Framework for Inference-time Scaling and Steering of Diffusion Models." Forty-second International Conference on Machine Learning, 2025.*

*[2] Kim, Sunwoo, Minkyu Kim, and Dongmin Park. "Test-time Alignment of Diffusion Models without Reward Over-optimization." The Thirteenth International Conference on Learning Representations, 2025.*

*[3] Ma, Nanye, et al. "Inference-time scaling for diffusion models beyond scaling denoising steps." arXiv preprint arXiv:2501.09732 (2025).*

*[4] Li, Xiner, et al. "Dynamic Search for Inference-Time Alignment in Diffusion Models." arXiv preprint arXiv:2503.02039 (2025).*

*[7] Jain, Vineet, et al. "Diffusion Tree Sampling: Scalable inference-time alignment of diffusion models." The Thirty-ninth Annual Conference on Neural Information Processing Systems, 2025.*

*[6] Dang, Meihua, et al. "Inference-time scaling of diffusion language models with particle gibbs sampling." arXiv preprint arXiv:2507.08390 (2025).*

**Questions:**

- In Figure 3, why is DFS-0.9 slightly lower than DFS-0.7 for the same amount of compute? Intuitively, it seems that thresholds should generally provide more high quality samples.
- In Figure 4 why why is best-of-N with 8 local steps worse across compute than BoN with 6 local steps?
- How do the BFS/DFS + local search compare with existing methods in terms of wall-clock times?

---

> ### Author Response · Authors · 2025-11-18
> **Reply to Reviewer TXqE (1/3)**
>
> We thank the reviewer for the detailed feedback, and we have revised the statements in our paper. Here we provide additional discussions and results.
>
> > The global search methods BFS/DFS proposed in the paper largely seem to be an alternate perspective or modifications to existing methods, rather than fundamentally new algorithms
>
> We acknowledge that the design space of BFS largely builds on prior work, and that our BFS experiments primarily serve to strengthen baseline methods and clarify existing design choices.
>
> For DFS, we emphasize that it is designed for adaptive backtracking to enable efficient global exploration, which is fundamentally different from the fixed schedules used in prior methods.
>
> We remark that the core contribution of our work is the introduction of a **unified framework for jointly scaling local and global search**. We show that integrating both components is essential across a variety of challenging decision-making domains. This principled and theoretically grounded framework enhances the robustness and adaptivity of inference-scaling methods, leading to strong empirical gains in areas such as planning and offline reinforcement learning. By jointly scaling local and global search, our approach achieves substantial improvements over both our strengthened BFS baseline and prior methods.
>
> > Comparison with SoP
>
> We thank the reviewer for the suggestion, and we have revised the presentation in Line 273-274 (Page 6). We have added discussions about SoP in related works (Page 3) and Section 4.1.2 (Page 6), and experiments in Table 2 (Page 7) and Figure 8 (Page 30).
>
>
> The key difference between DFS and SoP lies in the backtracking strategy. DFS employs an adaptive backtracking schedule, whereas SoP perturbs all particles with noise and then denoises them jointly, which can introduce substantial computational inefficiency. Moreover, DFS’s adaptive schedule enables early backtracking with restarts from T=0, a feature that is crucial for global exploration and for escaping low-quality modes.
>
> We compare SoP with both BFS and DFS. The BFS results are presented in the Table 2 (Page 7) with relevant information shown below. We also present the DFS results in compbench in [https://postimg.cc/DWnSRbsx](https://postimg.cc/DWnSRbsx) (Figure 8, Page 30)
>
> **SoP experiment results**
> | Model | N    | BoN | SoP | BFS  |
> | ----- | ---- | --- | --- | ---- |
> | SD v1.5      | 4     |   0.702±0.057  |  0.688±0.024   |    **0.882±0.029**  |
> | SD v1.5    |   8   |   0.896±0.031  |  0.884±0.022   |   **1.087±0.031**   |
> | SD XL      |   4   | 1.085±0.013    |  1.076±0.014   |   **1.194±0.024**   |
> | SD XL | 8 |  1.198±0.021   |  1.185±0.012   |**1.291±0.018**|
>
>
>
> Empirically, we find that SoP underperforms BoN in terms of global exploration efficiency, consistent with the results reported in Table 2 of [1]. We hypothesize that this is because SoP allocates equal compute to all particles during both noising and denoising, thereby reducing compute efficiency. In contrast, DFS demonstrates superior adaptivity and efficiency compared with prior methods.
>
> -----
>
>
> [1] Ma, Nanye, et al. "Inference-time scaling for diffusion models beyond scaling denoising steps." arXiv preprint arXiv:2501.09732 (2025).

---

> ### Author Response · Authors · 2025-11-18
> **Reply to Reviewer TXqE (2/3)**
>
> > DFS results
>
> Thank you for the helpful feedback. We have revised the description and added the base model scores accordingly.
>
> > Hyperparameter burden
>
> For the threshold parameter in DFS, our experiments in Fig. 3 (left) and Fig. 9 (Page 31) demonstrate that the method is robust across a wide range of practical threshold values.
>
>
> As discussed in Lines 1390–1392 (Page 26), the threshold can either be set to increase progressively during the denoising process, as in our Maze experiments, or remain constant, as in our Image experiments. In the Maze experiments, since the generated trajectories are coarse and low-quality at early evaluation timesteps, we adopt a lower threshold at early steps only to reject clearly out-of-distribution samples. We then use a tighter threshold at lower noise levels to enforce a stronger constraint. For the text-to-image experiments, since the reward scale remains similar across evaluation timesteps, we simply use a fixed threshold.
>
> We provide additional ablations of DFS in [https://postimg.cc/y3RTNZNt](https://postimg.cc/y3RTNZNt) and [https://postimg.cc/qNPnHdzk](https://postimg.cc/qNPnHdzk) （Figure 9, Page 31), demonstrating the robustness of our method.
>
> For the initial choice of $\delta$, we recommend practitioners start with the median of the reward distribution over generated samples, which is approximately 0.7 in our text-to-image experiments, and gradually increase the threshold as the compute budget grows to achieve better performance.
>
> > Insufficient experiment details in the main paper.
>
> We apologize for the earlier confusion regarding the experimental details. These details have now been moved to the main text in the revised version.
>
> For the prompt dataset, as noted in the caption of Figure 1 in [2], the ImageReward and HPS scores are computed using prompts from the ImageReward paper. We directly adopt the same prompt file from the official repository to ensure consistency.
>
> The number of recurrent steps used in the offline RL experiments is provided in Table 11 (Page 32).
>
> > Misunderstandings of the BFS framework:  SoP seems closer to DFS, and DSearch does not map trivially to the BFS framework.
>
> For SoP [1], we note that while SoP introduces noise injection similar in spirit to the backtracking mechanism of DFS, it operates under a fixed denoising schedule, which fundamentally differs from the adaptive backtracking used in DFS-style methods. We have added discussions and experiments accordingly.
>
> For DSearch [3], we apologize for the lack of discussion in our initial draft. We have added corresponding discussions in Section D.1.2 (Page 26) and included results in Table 2 (Page 7). Relevant results are also shown below:
>
> **DSearch experiments**
> | Model | N    | BoN | DSearch | BFS  |
> | ----- | ---- | --- | --- | ---- |
> | SD v1.5      | 4     |   0.702±0.057  | 0.836±0.032   |    **0.882±0.029**  |
> | SD v1.5    |   8   |   0.896±0.031  | 1.011±0.019   |   **1.087±0.031**   |
> | SD XL      |   4   | 1.085±0.013    |  1.139±0.018   |   **1.194±0.024**   |
> | SD XL | 8 |  1.198±0.021   |  1.252±0.019   |**1.291±0.018**|
>
> The branching process in DSearch can be interpreted as a two-phase beam search: (1) selecting the best sample among the M children of each node, and (2) selecting a subset of b nodes across all candidates. As shown in our experiments, DSearch exhibits lower computational efficiency compared to several prior methods. We hypothesize that this inefficiency arises from sampling the same M number of children for all particles in step (1), thereby allocating excessive computation to low-quality nodes.
>
>
> -----
>
>
> [1] Ma, Nanye, et al. "Inference-time scaling for diffusion models beyond scaling denoising steps." arXiv preprint arXiv:2501.09732 (2025).
>
> [2] Singhal, Raghav, et al. "A general framework for inference-time scaling and steering of diffusion models." arXiv preprint arXiv:2501.06848 (2025).
>
> [3] Li, Xiner, et al. "Dynamic Search for Inference-Time Alignment in Diffusion Models." arXiv preprint arXiv:2503.02039 (2025).

---

> ### Author Response · Authors · 2025-11-18
> **Reply to Reviewer TXqE (3/3)**
>
> > The wording in Section 3.2 is a bit misleading - it seems to suggest simply adding the score functions is enough to sample from product distributions.
>
> Thanks for pointing out! We have revised accordingly.
>
> > Statement about the scoring method using $x_{0|t}$ to account for variations during sampling
>
> Here, we score each particle not only by its posterior reward mean $x_{0|t}$ at the the current timestep $t$, but also by incorporating its past reward estimates : ${x}_{0|s}, s>t$, thereby accounting for reward variations throughout the sampling process. We have clarified this representation in the revised version.
>
> > Discussions about more relevant papers
>
> Thank you for pointing out these relevant papers. We have added these discussions accordingly in related works (Page 3).
>
> > Why is DFS-0.9 slightly lower than DFS-0.7
>
> A higher threshold such as 0.9 can lead DFS to backtrack more significantly on saturated instances. While this might secure marginally better solutions, the resulting reduction in compute efficiency slightly worsens the Pareto point compared to DFS-0.7. However, these higher thresholds demonstrate better scaling with high compute budgets; they continue to improve performance with increased compute, whereas lower thresholds tend to encounter performance and compute saturation earlier.
>
> > Why is best-of-N with 8 local steps worse across compute than BoN with 6 local steps?
>
> Compute is measured as the total cost of both local and global search. For the same compute budget, best-of-N with 8 local steps corresponds to fewer particles than best-of-N with 6 local steps. Since local search alone cannot effectively explore multi-modal landscapes or transition between modes, its marginal improvements decrease when global search is reduced.
>
> > How do the BFS/DFS + local search compare with existing methods in terms of wall-clock times?
>
> We have added a wall-clock time analysis in Section F, Table 12 (Page 34), where we evaluate the average time to generate a sample in the class-guided ImageNet generation task on a single NVIDIA RTX 4090 GPU. Relevant results are shown below.
>
> **Wall-clock time for class-guided ImageNet generation**
> | Method       | Settings                                  | Compute(NFEs) | Seconds |
> | ------------ | ----------------------------------------- | ------------- | ------- |
> |        Direct sampling      |  N=1                                         |    100           |    14     |
> |       DAS       |      N=1, 1 gradient step    |   100            |    26     |
> |    Local search          |  N=1, 1 additional local search step                                         |  200             |  40      |
> | Local search | N=1, 2 additional local search step       | 300           | 66      |
> | BoN          | N=4                                       | 400           | 55     |
> | BFS          | N=4                                       | 400           | 56      |
> | DAS          | N=4, 1 gradient step                      | 400           | 106      |
> | DFS          | budget=4                                  | 210           | 29      |
> | BoN          | N=4, 2 additional local search steps      | 1200          | 260     |
> | BFS          | N=4, 2 additional local search steps      | 1200          | 261     |
> | DFS          | budget=4, 2 additional local search steps | 620           | 136     |
>
> The results show that wall-clock time is dominated primarily by NFEs, and the sparse verifier evaluations used in global search introduce only marginal overhead. Gradient-based methods are slower due to the additional gradient computations, and DFS is slightly slower as well because it generates particles sequentially, limiting the ability to fully leverage GPU batch parallelism.
>
>
> -----
>
>
> [1] Ma, Nanye, et al. "Inference-time scaling for diffusion models beyond scaling denoising steps." arXiv preprint arXiv:2501.09732 (2025).
>
> [2] Singhal, Raghav, et al. "A general framework for inference-time scaling and steering of diffusion models." arXiv preprint arXiv:2501.06848 (2025).
>
> [3] Li, Xiner, et al. "Dynamic Search for Inference-Time Alignment in Diffusion Models." arXiv preprint arXiv:2503.02039 (2025).

---

> ### Comment · Reviewer_TXqE · 2025-11-23
>
> Thank you for your detailed response. The additional comparisons, more details in the paper, and clarifications on statements are much appreciated. I have some outstanding concerns/questions which I detail below:
>
> - Thank you for clarifying the contributions of this work. A large part of the paper focuses on explanation and empirical analysis of BFS/DFS frameworks, hence the message that the core contribution was jointly scaling local and global search became somewhat diluted. Out of the contributions listed in the introduction, point 1 refers to BFS/DFS (analysis or tweaks of existing methods), point 2 refers to annealed Langevin MCMC (a known technique), and only point 3 mentions jointly scaling local and global search.
> Based on the authors’ response, my understanding of the joint local + global search is that previous works (TDS, DAS) have proposed using a gradient-guided diffusion process (which can be seen as equivalent to single step of local search) along with guided sampling methods like SMC, this work suggests using *multiple* recurrent steps for local search, and combining them with largely existing methods (BFS/DFS/BoN) for more effective test-time scaling. Is this understanding correct?
>
> - The discussion in the related works section is appreciated, however, I think the introduction also needs to be updated to correctly position this work relative to these prior works. Currently, the introduction (line 78) asserts that “tree-based methods (Guo et al., 2025), typically as BFS with fixed schedules”. In light of works like [Jain et al. 2025, Lee et al. 2025], this statement should be revised since these methods are quite different from BFS and more importantly, both works enable adaptive allocation of compute at test-time by using search methods.
> - The response to reviewer JCZg has clarified that the DAS baseline in the paper did not use gradient-guidance for the proposal, which is a very important detail and should have been mentioned clearly in the paper. I appreciate the authors adding this detail during the rebuttal stage.
> To avoid potential misunderstandings - I suggest the authors rename their DAS baseline in Table 2, Figure 4, and elsewhere to clearly highlight that this is not the original method (especially since Table 2 specifically cites Kim et al., 2025 despite it not being the same method).
> - Regarding the offline RL experiments, Table 11 shows that TTS uses one local search step for a vast majority of the datasets. Since this should be essentially the same as the gradient-guided proposal used by DAS, could the authors explain why TTS (BoN + 1 step gradient guidance) outperforms DAS (SMC + 1 step gradient guidance)? If this is DAS without the gradient term, then I suggest the authors rename the method as indicated above and add a comparison with DAS as originally proposed.

---

> > ### Author Response · Authors · 2025-11-24
> > **Reply to Reviewer TXqE (1/2)**
> >
> > Thank you for your detailed follow-up. We address your concerns point by point below.
> >
> > > Out of the contributions listed in the introduction, point 1 refers to BFS/DFS (analysis or tweaks of existing methods), point 2 refers to annealed Langevin MCMC (a known technique), and only point 3 mentions jointly scaling local and global search. Based on the authors’ response, my understanding of the joint local + global search is that previous works (TDS, DAS) have proposed using a gradient-guided diffusion process (which can be seen as equivalent to single step of local search) along with guided sampling methods like SMC, this work suggests using multiple recurrent steps for local search, and combining them with largely existing methods (BFS/DFS/BoN) for more effective test-time scaling.
> >
> > We provide the following clarifications regarding our contributions.
> >
> > 1. Although BFS is related to particle-based and tree-search methods, our adaptive DFS is fundamentally different from prior approaches (e.g., DAS, FK, TreeG, DSearch) because it supports adaptive compute allocation for each instance. This allows the method to focus computation where it is most useful, which we find leads to consistently stronger performance in our experiments. Compared with SoP, which uses a similar noise-injection idea, our adaptive schedule performs significantly better, underscoring the importance of adaptivity in inference scaling.
> >
> >
> > 2. While annealed Langevin MCMC is a known technique, its connection to diffusion-model guidance has not been clearly established in prior work. We are, to the best of our knowledge, the first to formalize the equivalence between recurrent training-free guidance and Langevin MCMC. This provides a clearer theoretical understanding of our approach and helps unify earlier guidance methods.
> >
> >     We also note that Langevin MCMC is different from one-step gradient guidance. One-step guidance relies on guided diffusion sampling, but directly sampling from the posterior through reverse diffusion is intractable. In contrast, annealed Langevin MCMC provides asymptotically exact sampling, which is what we adopt in our framework. The recurrent steps act as Langevin MCMC applied to the original data distribution, helping prevent out-of-distribution drift across multiple gradient updates, and are essential for successfully scaling local search.
> >
> > 3. We combine local search with global search because each one alone has limitations. Local search struggles to explore all the modes of the target distribution, and global search has difficulty producing refined samples in low-probability regions. Using them together addresses both issues. This combined framework also naturally leads to our double-verifier method, which helps reduce reward hacking in gradient-based approaches.

---

> ### Author Response · Authors · 2025-11-24
> **Reply to Reviewer TXqE (2/2)**
>
> > I think the introduction also needs to be updated to correctly position this work relative to these prior works.  In light of works like [Jain et al. 2025, Lee et al. 2025], this statement should be revised since these methods are quite different from BFS and more importantly, both works enable adaptive allocation of compute at test-time by using search methods.
>
> Thank you for the suggestion. We have updated the introduction accordingly and added the relevant references (Line 78-85). For the concurrent works by Jain et al. and Lee et al., we have included a more detailed discussion, highlighting their approaches to adaptive compute allocation (Line 117-122).
>
> > To avoid potential misunderstandings - I suggest the authors rename their DAS baseline in Table 2, Figure 4, and elsewhere to clearly highlight that this is not the original method
>
> We have updated the BFS section to rename DAS as “DAS (w/o grad)”, and this change is reflected in Table 2 and Table 7, and in the text Line 271, Line 346-347, and Line 372. For Figure 2 in the Maze experiment, because our local search uses gradient guidance, we also used the original DAS version with gradients to ensure a fair comparison.
>
> > Table 11 shows that TTS uses one local search step for a vast majority of the datasets. Since this should be essentially the same as the gradient-guided proposal used by DAS, could the authors explain why TTS (BoN + 1 step gradient guidance) outperforms DAS (SMC + 1 step gradient guidance)?
>
> Your observation is correct: the DAS we used is the original version with gradients. There are two main reasons why TTS outperforms DAS.
>
>
> First, for most tasks with only one local search step, DAS performs similarly to TTS, and in some cases even slightly better (e.g., Hopper Medium-Expert). **Importantly, the tasks that DAS clearly underperforms (Hopper Medium, Walker2d Medium-Replay) are precisely the tasks where TTS adopts multiple local search steps.** This highlights the value of scaling up local search.
>
>
> Second, in TTS, because recurrent training-free guidance is equivalent to annealed Langevin MCMC, we obtain a unified parameterization for gradient guidance that covers both guidance on $x\_t$ and $x\_{0|t}$. We can also separate the guidance term from the resampling function. As a result, even with the same single local search step, TTS can slightly outperform DAS on some tasks. This shows the advantage of having a unified framework.

---

> > ### Comment · Reviewer_TXqE · 2025-11-24
> >
> > Thank you for the response. With the explanation about the offline RL results, changes to the introduction, and the clarifications regarding contributions, my concerns are largely addressed. I will wait for additional discussions with other reviewers before updating my review.

---

> > > ### Author Response · Authors · 2025-11-24
> > >
> > > Thank you for your prompt reply and constructive review! We’re happy to discuss any further questions you may have.

---

### Official Review · Reviewer_KEXB · 2025-11-04

**Soundness:** 3
**Presentation:** 4
**Contribution:** 3
**Rating:** 8
**Confidence:** 3

**Summary:**

The paper proposes a unified inference-time scaling framework for diffusion models that performs global exploration with BFS/DFS over the denoising tree and local refinement via annealed Langevin MCMC guided by a verifier. It clarifies and improves BFS-style methods (tempering, scoring, SSP resampling) and introduces the first adaptive DFS with verifier-threshold backtracking to allocate compute on demand. It demonstrates superior Pareto efficiency across image synthesis, planning, and offline RL, including policy distillation and a double-verifier to curb reward hacking.

**Strengths:**

- Principled, general framework unifying global tree search and local gradient-based MCMC; theory linking recurrence to Langevin MCMC with convergence insights
- Algorithmic contributions: improved BFS design and novel adaptive DFS that scales compute by instance difficulty, substantially boosting quality per NFE
- Strong empirical results and ablations across tasks; new Pareto frontier, competitive offline RL without retraining, effective policy distillation, and robustness via double-verifier

**Weaknesses:**

- The proposed BFS and DFS methods were shown to be effective across various scenarios. However, establishing a unified methodology with default hyperparameter settings (beyond TTS in RL tasks) would make the approach more practical for broader use.
- No comparisons with more recent T2I diffusion models beyond SD1.5 and SDXL and recent baselines (e.g., [1]).
- How stable is the proposed method against reward hacking without the double verifier? Since reward hacking is a critical issue, it should be clarified whether the double verifier is essential for robustness.

---
[1] Inference-time scaling for diffusion models beyond scaling denoising steps, CVPR, 2025

**Questions:**

- If the gradient-based guidance component were removed, could the proposed method generalize to discrete diffusion (e.g., MaskGIT, LLaDA)? Any results or insights on this would be valuable.
- Could this approach be extended to Flow models as well?

---

> ### Author Response · Authors · 2025-11-18
> **Reply to Reviewer KEXB**
>
> We thank the reviewer for the thoughtful suggestions, and we provide discussions and additional results below.
>
> > Unified design space for BFS and DFS
>
> We agree that developing a more principled design space for global search methods such as BFS and DFS would be valuable. In our current setup, the DFS threshold and BFS temperature serve as control knobs that balance the exploration–exploitation tradeoff. For backtracking depth we adopt simple heuristics such as using T/2 or T/4, where T is the total number of denoising steps. It is also possible to use reward statistics, such as choosing the threshold as the median reward, which may facilitate parameter transfer across different settings and verifiers with different reward scales.
>
> > Comparison with recent diffusion models and SoP
>
> We have added these experiments in Table 2 (Page 7), with relevant information shown below:
>
>
> **Result for BFS-style global search with FLUX.1 dev**
> |    Model | N | BoN        |     FK        |  DAS           |   TreeG          |   SVDD          |   SoP          |    DSearch         |           BFS  |
> | --- | ----- | ----------- | ----------- | ----------- | ----------- | ----------- | ----------- | ----------- | ----------- |
> |  FLUX.1 dev   |   4    | 1.113±0.015 | 1.145±0.017 | 1.194±0.013 | 1.178±0.018 | 1.069±0.035 | 1.104±0.011 | 1.169±0.022     | **1.203±0.013** |
>
> We observe that SoP underperforms the particle-based baselines, which we hypothesize is due to compute inefficiency arising from its uniform noise injection and denoising schedules for all particles.
>
>
> > Robustness of our method against reward hacking
>
> In the text-to-image experiments, the HPSv2 metric in Table 7 (Page 28) and visualizations in Figures 6 (Page 28) and 7 (Page 29) show that inference scaling primarily enhances the prompt-relevant semantics while preserving aesthetic quality, without exhibiting signs of reward hacking.
>
> For the decision-making tasks such as Maze and offline RL, we emphasize that these tasks are evaluated using ground-truth rewards. Our superior results in these settings further demonstrate the robustness of our method.
>
> > Extension to discrete diffusion models
>
> Since discrete diffusion sampling exhibits a similar tree structure, principled tree-search methods such as BFS and DFS can also be applied. Although gradient-based Langevin MCMC is intractable in this setting, we sometimes have access to exact likelihood or energy functions derived from the logits, which enables the use of Metropolis–Hastings methods. Additionally, recurrence can be similarly incorporated into the sampling process by iteratively masking and unmasking tokens between adjacent mask ratios.
>
> > Extension to flow models
>
> Yes, our method can be readily extended to flow models. For global search, BFS and DFS can be directly applied to the tree-structured sampling process of flow models. To enable stochatic sampling in flow models, we can apply the stochastic samplers in [1].
>
> For local search, we provide an additional proof in Section C.3.1 for the flow-matching noise schedule and sampler, showing that recurrence in flow models is also equivalent to Langevin MCMC. This establishes that our local search procedure applies equally well to flow-based models.
>
> ---
>
> [1] Wang, Feng, and Zihao Yu. "Coefficients-Preserving Sampling for Reinforcement Learning with Flow Matching." arXiv preprint arXiv:2509.05952 (2025).

---

### Meta-Review · Area_Chair_Bj4w · 2025-12-30

**Summary:**

This work proposes a unified framework for inference-time scaling via global tree search and local annealed Langevin MCMC. The submission presents a principled approach that establishes a new Pareto frontier across planning, offline RL, and image generation tasks. Reviewers initially questioned the novelty and expressed doubts due to the complexity of the hyperparameters, but many of the concerns were addressed in the rebuttal. The authors demonstrated that their adaptive DFS has better efficiency compared to SoP and DSearch baselines and validated robustness against reward hacking through double-verifier experiments and offline RL analysis. The reviewer acknowledged the rebuttal, and their concerns were largely addressed, suggesting a consensus for acceptance. ("Suggesting" only, since Reviewer TXqE was waiting for the reviewers' discussion phase to update their score, but that did nit occur.)

I recommend **acceptance as a poster.**

**Reviewer Concerns:**

The main concerns reported in the reviews are the following:
- **Novelty**: Critiques regarding the similarity to existing methods like FK-Steering and SoP were addressed by clarifying the differences in adaptivity of the proposed DFS (Reviewers TXqE and JCZg) and adding requested comparisons to SoP and DSearch (Reviewer KEXB).
- **Hyperparameter complexity**: Concerns regarding the hyperparameter tuning and fixed backtracking schedules (Reviewers Vc1j and TXqE) were resolved through additional ablation studies showing robustness and providing new heuristics for parameter selection.
- **Reward hacking**: Concerns about reward over-optimisation (Reviewers KEXB and JCZg) were mitigated by introducing double-verifier experiments and reporting HPSv2 and FID metrics to ensure sample quality.

**Reviewer Scores:**

- **Vc1j: Rating: 8 / Confidence: 4.** This reviewer explicitly confirmed that the additional ablation studies *"fully answered"* their concerns. They would likely maintain or increase their score of 8.
- **TXqE: Rating: 2 / Confidence: 4.** After the authors clarified the novelty and addressed the DAS baseline comparison, this reviewer explicitly stated that their *"concerns are largely addressed"* but they were waiting for the reviewers' discussion phase. They would probably have raised their score to a 6.
- **KEXB: Rating: 8 / Confidence: 3.** The authors provided the requested additional comparisons with recent diffusion models and SoP. They would likely maintain or increase their score of 8.
- **JCZg: Rating: 6 / Confidence: 4.** The authors addressed the critiques regarding the justification for Langevin MCMC and provided additional metrics (HPS scores) to counter reward hacking concerns. This score would likely remain a 6 (or possibly slightly increase).

---

### Decision · Program_Chairs · 2026-01-26

Accept (Poster)